# Purine nucleosides replace cAMP in allosteric regulation of PKA in trypanosomatid pathogens

**Veronica Teresa Ober**[1†], **George Boniface Githure**[1†], **Yuri Volpato Santos**[1†], **Sidney Becker**[2,3], **Gabriel Moya Munoz**[1], **Jérôme Basquin**[4], **Frank Schwede**[5], **Esben Lorentzen**[6], **Michael Boshart**[1]*

[1]Faculty of Biology, Genetics, Ludwig-Maximilians University Munich (LMU), Martinsried, Germany; [2]Max Planck Institute of Molecular Physiology, Dortmund, Germany; [3]TU Dortmund, Department of Chemistry and Chemical Biology, Dortmund, Germany; [4]Max Planck Institute for Biochemistry, Martinsried, Germany; [5]BIOLOG Life Science Institute GmbH & Co KG, Bremen, Germany; [6]Department of Molecular Biology and Genetics, Aarhus University, Aarhus, Denmark

**\*For correspondence:**
boshart@lmu.de

[†]These authors contributed equally to this work

## Abstract

Cyclic nucleotide binding domains (CNB) confer allosteric regulation by cAMP or cGMP to many signaling proteins, including PKA and PKG. PKA of phylogenetically distant *Trypanosoma* is the first exception as it is cyclic nucleotide-independent and responsive to nucleoside analogues (Bachmaier et al., 2019). Here, we show that natural nucleosides inosine, guanosine and adenosine are nanomolar affinity CNB ligands and activators of PKA orthologs of the important tropical pathogens *Trypanosoma brucei*, *Trypanosoma cruzi*, and *Leishmania*. The sequence and structural determinants of binding affinity, -specificity and kinase activation of PKAR were established by structure-activity relationship (SAR) analysis, co-crystal structures and mutagenesis. Substitution of two to three amino acids in the binding sites is sufficient for conversion of CNB domains from nucleoside to cyclic nucleotide specificity. In addition, a trypanosomatid-specific C-terminal helix (αD) is required for high affinity binding to CNB-B. The αD helix functions as a lid of the binding site that shields ligands from solvent. Selectivity of guanosine for CNB-B and of adenosine for CNB-A results in synergistic kinase activation at low nanomolar concentration. PKA pulldown from rapid lysis establishes guanosine as the predominant ligand in vivo in *T. brucei* bloodstream forms, whereas guanosine and adenosine seem to synergize in the procyclic developmental stage in the insect vector. We discuss the versatile use of CNB domains in evolution and recruitment of PKA for novel nucleoside-mediated signaling.

## eLife assessment

This **landmark** study sheds light on a long-standing puzzle in Protein kinase A activation in Trypanosoma. Extensive experimental work provides **exceptional** evidence for the conclusions of the work, which represents a significant advancement in our understanding of the molecular mechanism of cyclic nucleotide binding domains. The work is relevant for researchers with interests in kinases and their mechanistic study.

## Introduction

Protein kinase A (PKA) is a prototype kinase first purified from rabbit skeletal muscle in 1968 (***Walsh et al., 1968***). More than 40 years of trailblazing biochemical and structural work elucidated the

mechanism of allosteric activation by cAMP, providing a paradigm of allosteric regulation (*Taylor et al., 2021*). Inactive PKA is a dimeric or tetrameric complex of regulatory (R) and catalytic (C) subunits, depending on the species. Upon activation, two molecules of cAMP bind to two cyclic nucleotide binding domains (CNB) arranged in tandem in the C-terminal part of the regulatory subunit(s). Cyclic AMP binding to the C-terminal CNB-B initiates a conformation change that opens up the adjacent CNB-A for a second cAMP molecule whose binding completes the conformational transition that liberates the C-subunit from the holoenzyme complex (*Kim et al., 2007*). The free C subunit is thereby released from autoinhibition and activated (*Kim et al., 2007*; *Taylor et al., 2012*). PKA is a highly conserved kinase present in all eukaryotes except plants, functioning in diverse signaling processes ranging from metabolic regulation and hormone action to cell differentiation and synaptic long-term potentiation underlying memory (*Lee et al., 2021*). In protists and fungi, the predominant functions are response to carbon source changes and regulation of developmental transitions, infectivity, or sexual dimorphism (*Perrin et al., 2020*; *Hitz et al., 2021*; *Uboldi et al., 2018*; *Jia et al., 2017*; *Choi et al., 2015*; *Kim et al., 2021*; *Vaidyanathan et al., 2014*). Regulation of PKA by cAMP was universally found (*Rinaldi et al., 2010*; *Haste et al., 2012*; *Kurokawa et al., 2011*; *Taylor et al., 2012*), which is why PKA and cAMP-dependent protein kinase are used as synonyms. Furthermore, cAMP dependence was assumed and widely cited in reviews for PKAs of species for which uncontested biochemical evidence is lacking, including protozoan flagellates of the phylogenetically distant order *Kinetoplastida*. It was therefore of greatest interest that PKA in *Trypanosoma brucei*, a parasitic and pathogenic member of the *Kinetoplastida* was found unresponsive to cAMP even at high intracellular concentrations (*Bachmaier et al., 2019*; *Bubis et al., 2018*). Compound screening then identified 7-deazapurine nucleoside antibiotics as nanomolar activators of PKA in *T. brucei* (*Bachmaier et al., 2019*).

*T. brucei*, *Trypanosoma cruzi*, and *Leishmania* sp. are related trypanosomatid pathogens causing the deadly neglected tropical diseases sleeping sickness, Chagas disease and leishmaniosis, respectively. In addition, *Trypanosoma* is responsible as animal pathogen for important economic losses and impedes social development in affected countries. These organisms are famous for discovery of many exotic biochemical and genetic mechanisms (*Matthews, 2015*), and not surprisingly, signaling systems diverge from other model organisms and few pathways are on the way to be assembled (*Matthews, 2021*). Nevertheless, cAMP seems to play an important role: *T. brucei* encodes >80 adenylate cyclase genes (*Salmon et al., 2012a*) that are important for host innate immunity subversion (*Salmon et al., 2012b*) and for development of the parasite in its insect vector (*Bachmaier et al., 2022*). These pathways are obviously uncoupled from the cAMP unresponsive PKA and seem to use alternative and novel cAMP effectors (*Bachmaier et al., 2023*). PKA in these parasites is essential and important for cell division (*Bachmaier et al., 2019*; *Baker et al., 2021*; *Cayla et al., 2022*). In *T. brucei*, PKA has been identified as candidate member of a quorum sensing pathway and implicated in stage development (*Mony et al., 2014*; *Toh et al., 2021*). The genome of *T. brucei* encodes one regulatory and three catalytic subunits that all have syntenic orthologs in *T. cruzi* and *Leishmania*. The upstream pathway(s) regulating PKA in these organisms have not been identified. However, the high degree of conservation of the CNBs in TbPKA and their high affinity binding to nucleoside analogues (*Bachmaier et al., 2019*) suggested the existence of an alternative second messenger. PKG, a related AGC kinase, is also subject to allosteric regulation by cyclic nucleotides, responding to cyclic guanosine monophosphate (cGMP) instead of cAMP (*Huang et al., 2014b*; *Kim and Sharma, 2021*). The high structural similarity of the CNB domains in PKA and PKG spurred attempts to define determinants of cyclic nucleotide binding selectivity. Amino acids that contribute to selectivity have been identified (*Lorenz et al., 2017a*; *Shabb et al., 1991*; *Shabb et al., 1990*; *Huang et al., 2014a*; *Weber et al., 1989*; *Corbin et al., 1986*; *Kim and Sharma, 2021*; *Lorenz et al., 2017a*), but establishing a consensus of key determinants has been challenging. Differences in ligand specificity of PKAR between a pathogen and its host provides an opportunity for much needed drug development to fight neglected vector borne diseases caused by *Trypanosomatida*. Hence, identification of physiological ligands of trypanosomatid PKA and definition of their binding selectivity were important goals.

Here, we show that purine nucleosides exhibit nanomolar affinity for the PKA regulatory subunits of these pathogens and activate the kinases. We define the minimal changes that convert a nucleoside-specific CNB to cyclic nucleotide specificity. Furthermore, we see site-selective binding and synergy between guanosine and adenosine, compatible with binding in vivo to PKAR. Expansion of the ligand

portfolio of CNBs in evolution has enabled repurposing of PKA for a different signaling pathway, while maintaining the sophisticated allosteric activation mechanism triggered by ligand binding to PKA.

## Results

### Nucleosides are direct activators of PKA in trypanosomes

We recently identified the nucleoside analogue 7-cyano-7-deaza-inosine (7-CN-7-C-Ino, Jaspamycin) and related compounds like Toyocamycin as potent activators of the cAMP-independent PKA of *Trypanosoma* (*Bachmaier et al., 2019*). Attempts to bioinformatically detect pathways for synthesis of these nucleoside antibiotics in trypanosomatids have been unsuccessful. Therefore, we considered unmodified purine nucleosides and studied the structure-activity-relationship (SAR) for kinase activation (*Figure 1a* and *Table 1*). Tagged R- and C1-subunits of *T. brucei* PKA were co-expressed in *Leishmania tarentolae* and stoichiometric holoenzyme complexes were tandem-affinity purified (*Figure 1—figure supplement 1a*). The tandem-affinity purification to near homogeneity guaranteed removal of any heterologous complexes formed with endogenous PKA subunits of the expression system. $EC_{50}$ values for kinase activation were determined from dose response assays (*Figure 1b*, *Table 1*, *Figure 1—figure supplement 1a*). Surprisingly, inosine was the most potent activator ($EC_{50}$ 14 nM). We did not expect this result as the structure of TcPKAR bound to 7-CN-7-C-Ino (PDB: 6FTF) and computational docking of 7-deaza analogues had suggested an important role of the cyano group at position 7 of the purine ring (*Bachmaier et al., 2019*). The SAR analysis (*Figure 1a*, *Table 1*, *Figure 1—figure supplement 1a*) showed the oxygen at position 6 in the purine ring to be particularly important, as nebularine, lacking a 6-substitution, was 186-fold less potent than inosine. An amino group substitution at position 6 (adenosine) resulted in a further twofold drop in potency. An amino group at position 2 (guanosine) caused 11-fold and a keto group in this position (xanthosine) a >4400-fold lower activation potency, respectively. A structural isomer of inosine (allopurinol riboside) with restricted delocalized π-electron system showed 120-fold reduced activation. The structure of TcPKAR bound to 7-CN-7-C-Ino (*Bachmaier et al., 2019*) predicted an important role of the ribose moiety that is accommodated deep in the binding pocket. The 2′-, 3′- and 5′-deoxy derivatives of adenosine confirmed essential roles for all three hydroxyl groups of the ribose ring (*Table 1*, *Figure 1—figure supplement 1a*). Inosine and guanosine 5′-monophosphates were >7700-fold less potent than the respective nucleosides and AMP did not activate even at 5 mM. Cyclic GMP activated the kinase in the upper micromolar range, whereas cAMP and cIMP were inactive up to 5 mM. Pyrimidine nucleosides uridine and cytidine were 3–4 orders of magnitude less potent than purine nucleosides (*Table 1*, *Figure 1—figure supplement 1a*). In summary, the natural nucleoside inosine is only twofold less potent as activator of TbPKAR than the nucleoside analogue activator 7-CN-7-C-Ino (6.5 nM; *Bachmaier et al., 2019*) but fivefold more potent than cAMP activation of the recombinant mammalian PKARIα2-2PKACα holoenzyme purified from *Escherichia coli* (*Figure 1c*, *Table 1*). Therefore, purine nucleosides qualify as possible physiological activators of TbPKA in trypanosomes.

### Nucleoside activation of PKA in kinetoplastid pathogens

Next, we asked if activation by nucleosides and complete insensitivity to cAMP is a unique feature of *T. brucei* PKA or a shared feature in the protozoan class of Kinetoplastida. We selected the medically important *T. cruzi* and *Leishmania donovani* as representatives of this group. Orthologous regulatory subunits (TcPKAR and LdPKAR1) and catalytic subunits (TcPKAC2 and LdPKAC1), respectively, were tagged and co-expressed in *L. tarentolae*, and holoenzyme complexes were tandem affinity purified (*Figure 1—figure supplement 1b, c*). The kinase assay dose responses (*Figure 1c*, *Figure 1—figure supplement 1b, c,*) show that inosine is the most potent of the tested nucleosides for all analysed species, whereas no activation was observed with cAMP, even at very high concentrations (*Figure 1c*, *Table 1*). Activation potency of inosine or guanosine was between threefold and 23-fold lower for *Leishmania* and *T. cruzi*, compared to *T. brucei*. The mammalian RIα2:Cα2 holoenzyme, included as control, was activated by cAMP with an $EC_{50}$ of 75 nM in agreement with *Herberg et al., 1996*, but was completely insensitive to inosine. We conclude that the PKAs of *T. cruzi* and *L. donovani* are also cAMP-independent nucleoside activated kinases. The same order of potency was found among the three tested nucleosides, adenosine being the weakest activator ($EC_{50}$ ~6–8 μM) of PKA in the three parasite species (*Figure 1—figure supplement 1b–c*, *Table 1*).

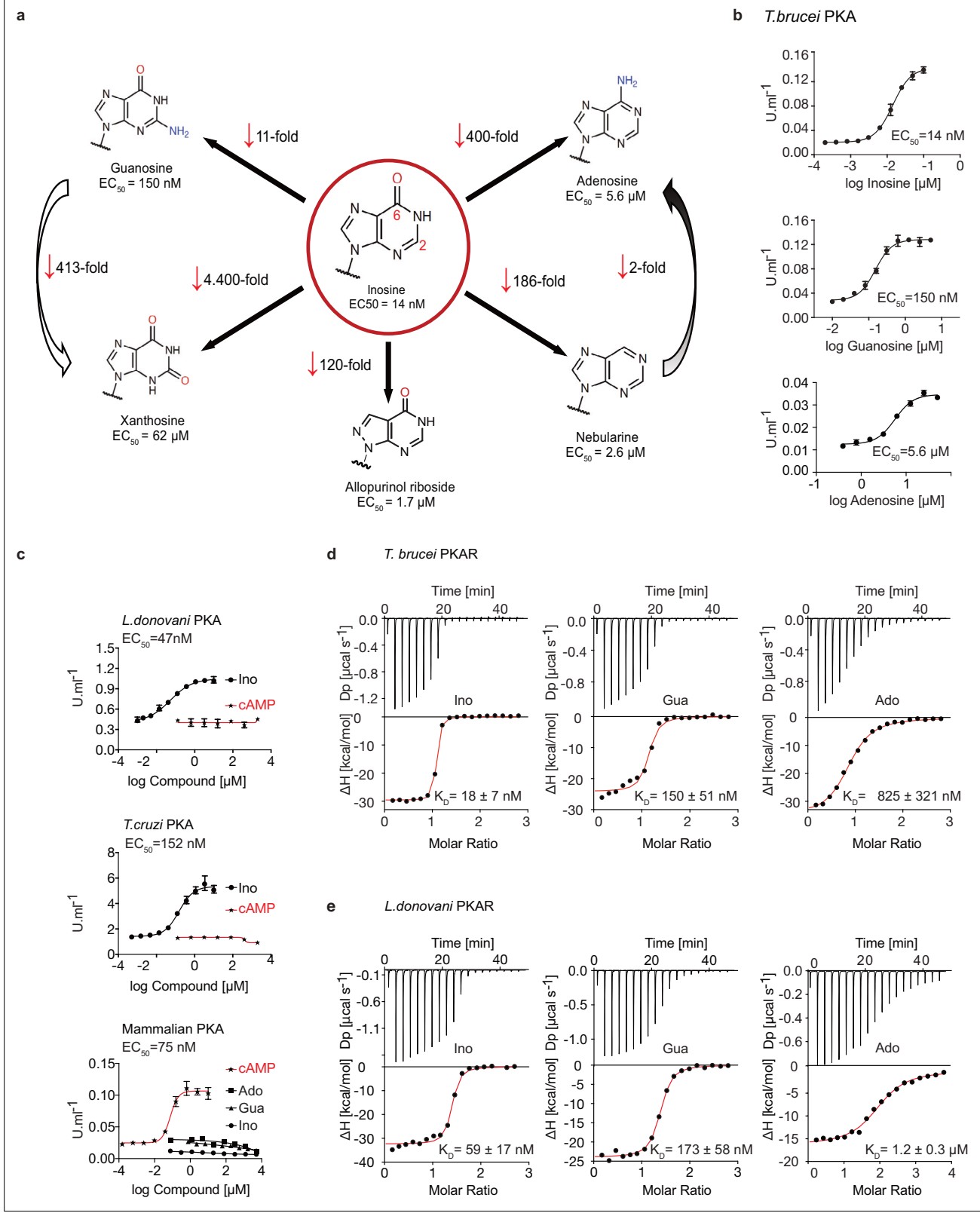

**Figure 1.** Trypanosomatid PKA binds to and is selectively activated by purine nucleosides. (**a**) Structure-activity relationship (SAR) analysis for TbPKA kinase activation by nucleoside derivatives. Chemical structures and the corresponding EC50 values are taken from *Table 1*. For representative dose-response curves see *Figure 1—figure supplement 1*. (**b**) Representative dose-response curves for activation of *T. brucei* PKAR-PKAC1 holoenzyme by inosine, guanosine or adenosine (in vitro kinase assay, n≥3 biological replicates). The calculated EC50 values are displayed next to the graph and in

*Figure 1 continued on next page*

*Figure 1 continued*

*Table 1*, error bars indicate SD of technical triplicates. Purity of PKA enzymes is shown in *Figure 1—figure supplement 1a*. (**c**) Representative dose-response curves for activation of *L. donovani*, *T. cruzi* and mammalian (human RIα/mouse Cα) holoenzymes by purine nucleosides and cAMP, as in A. The calculated EC$_{50}$ values are displayed next to the curve and in *Table 1*, error bars indicate SD of technical triplicates; Purity of PKA holoenzyme is shown in *Figure 1—figure supplement 1b, c*. (**d**) Binding isotherms (ITC) of nucleoside-depleted (APO) *T. brucei* PKAR(199-499) upon titration with purine nucleosides. The graphs give the difference power (DP) between the reference and sample cells upon ligand injection as a function of time (upper panel). In the lower panel, the total heat exchange per mole of injectant (integrated peak areas from upper panel) is plotted against the molar ratio of ligand to protein. A representative curve out of ≥3 independent replicates is shown. The final given K$_D$ (as in *Supplementary file 1*) was calculated as the mean (± SD) of at least three independent experiment (see source data files). For purity of R subunit eluted from SEC see *Figure 2—figure supplement 1a, b*. (**e**) Binding isotherms (ITC) of nucleoside-depleted (APO) *L. donovani* PKAR1(200–502) upon titration with purine nucleosides, as in D. Purity, aggregation state and thermal stability of protein sample prior to binding assays is shown in *Figure 2—figure supplement 1c, d*.

The online version of this article includes the following source data and figure supplement(s) for figure 1:

**Source data 1.** In vitro kinase activation assays of *T. brucei* PKAR-PKAC1 holoenzyme by inosine, guanosine, or adenosine.

**Source data 2.** In vitro kinase activation assays of *T.cruzi* PKAR-PKAC2, *L. donovani* PKAR1-PKAC1 by inosine or cAMP (no activation) and of *H. sapiens* PKARIα-PKAC holoenzyme by inosine, guanosine, and adenosine (no activation) or cAMP.

**Source data 3.** ITC measurements for inosine, guanosine or adenosine titrated to nucleoside-depleted (APO) *T. brucei* PKAR(199-499).

**Source data 4.** ITC measurements for inosine, guanosine or adenosine titrated to nucleoside-depleted (APO) *L. donovani* PKAR1(200–502).

**Figure supplement 1.** Kinetoplastid PKA activation.

**Figure supplement 1—source data 1.** Original Coomassie stained gels of tandem affinity purification of *T.cruzi* PKAR-PKAC2, *L. donovani* PKAR1-PKAC1and *T. brucei* PKAR-PKAC1.

**Figure supplement 1—source data 2.** Original Coomassie stained gels of tandem affinity purification of *T.cruzi* PKAR-PKAC2, *L. donovani* PKAR1-PKAC1and *T. brucei* PKAR-PKAC1 (labelled).

**Figure supplement 1—source data 3.** In vitro kinase activation assays of *T. brucei* PKAR-PKAC1 holoenzyme by compounds.

**Figure supplement 1—source data 4.** In vitro kinase activation assays of *L. donovani* PKAR1-PKAC1 by guanosine and adenosine.

**Figure supplement 1—source data 5.** In vitro kinase assays for activation of *T. cruzi* PKAR-PKAC2 by guanosine and adenosine.

**Figure supplement 1—source data 6.** ITC measurement of cAMP titrated to *H.sapiens* PKARIα or *L. donovani* PKAR1(200–502).

**Figure supplement 2.** Mass spectrometry identification of ligands bound to recombinant PKAR.

**Figure supplement 2—source data 1.** Original Coomassie stained gel.

**Figure supplement 2—source data 2.** Original Coomassie stained gel (labelled).

**Figure supplement 3.** Thermodynamic signatures of ligand binding from ITC experiments.

## Binding of nucleosides to kinetoplastid PKAs

To further investigate purine nucleoside-specific allosteric regulation of kinetoplastid PKAs, nucleoside binding parameters were determined for the isolated R-subunits. The N-terminally truncated PKARs of the respective species containing the two tandem CNBs were expressed in *E. coli* and purified. Initially, binding assays with natively purified PKAR were inconclusive and highly variable using several methods. We concluded that the purified PKAR was at least partially bound by ligands or metabolites derived from *E. coli*, similar to the mammalian PKAR subunit that tightly binds cAMP when purified from bacteria (*Buechler et al., 1993*). To confirm this directly, TbPKAR or HsPKARIα purified from *E. coli* were boiled to denature protein and were separated by centrifugation. Supernatants containing released ligands were collected and tested in kinase assays with purified holoenzyme as before (*Figure 1—figure supplement 2*). The supernatant of the boiled HsPKARIα fully activated the mammalian holoenzyme, but not the *T. brucei* holoenzyme, as would be expected for cAMP. In contrast, the supernatant from boiled TbPKAR fully activated the *T. brucei* holoenzyme but not the mammalian one (*Figure 1—figure supplement 2b*). In the HsPKARIα-derived supernatant only cAMP was detected by LC-MS (*Figure 1—figure supplement 2d*), whereas in the TbPKAR-derived supernatant nucleosides (predominantly inosine) were detected (*Figure 1—figure supplement 2c*). This experiment qualitatively showed tight binding of nucleosides to TbPKAR in *E. coli*. Subsequently, we routinely denatured the purified His-tagged regulatory subunits to remove any pre-bound ligands (see Materials and methods). Refolding conditions were optimized by a buffer screen and monitored by differential scanning fluorimetry (nanoDSF; *Niesen et al., 2007*) and size exclusion chromatography (*Figure 2—figure supplement 1a–d*). The thermal stability of proteins was determined by nanoDSF that records changes of the ratio of intrinsic fluorescence at two wavelengths (330 and

**Table 1.** Structure activity relation (SAR) analysis for PKA holoenzyme activation.

| PKA holoenzyme complex | Ligand | EC$_{50}$ (95% CI)* |
|---|---|---|
| *T. brucei* PKAR/PKAC1 | Inosine | 14 (13–15) nM |
| | Guanosine | 152 (132–172) nM |
| | Adenosine | 7.0 (6.9–8.4) μM |
| | cAMP | -† |
| | cGMP | 0.36 (0.33–0.41) mM |
| | cIMP | -† |
| | AMP | -† |
| | GMP | 1.1 (0.9–1.3) mM |
| | IMP | 108 (83–135) μM |
| | 2'-deoxyadenosine | -† |
| | 3'-deoxyadenosine | -† |
| | 5'-deoxyadenosine | -† |
| | Nebularine | 2.6 (2.2–3.2) μM |
| | Allopurinol riboside | 1.7 (1.5–1.9) μM |
| | Xanthosine | 62 (51–72) μM |
| | Uridine | 40 (35–47) μM |
| | Cytidine | ≥350 μM |
| *Leishmania* PKAR1/PKAC1 | Inosine | 47 (33–63) nM |
| | Guanosine | 1.7 (1.4–2.1) μM |
| | Adenosine | 6.5 (5.7–7.6) μM |
| | cAMP | -† |
| *T. cruzi* PKAR/PKAC2 | Inosine | 150 (110–200) nM |
| | Guanosine | 3.5 (2.8–4.5) μM |
| | Adenosine | 8.3 (5.2–12.4) μM |
| | cAMP | -† |
| human RIα/mouse Cα | Inosine | -† |
| | Guanosine | -† |
| | Adenosine | -† |
| | cAMP | 75 (59–93) nM |

*Mean half activation constants (EC$_{50}$) and 95% confidence interval (95% CI) determined from **Figure 1—figure supplement 1** using Graphpad prism 7.0 for technical triplicates.

†No activation was detected up to a maximum concentration of 5 mM.

350 nM). Natively purified TbPKAR(199-499) unfolded at T$_m$ of 59,5 °C. When refolded in absence of ligand (APO form) the T$_m$ was only 42,3 °C (**Figure 2—figure supplement 1b**). This is interpreted as stabilization of the purified TbPKAR by its partial loading with nucleosides from *E. coli*. Indeed, upon saturating the ligand-bound state of the refolded and the natively purified protein preparations by addition of excess inosine, the T$_m$ raised to 68 °C for both. The identical T$_m$ strongly indicates correct folding after renaturation. Ligand-depleted LdPKAR1(200–502) was prepared in the same way (**Figure 2—figure supplement 1c–d**), whereas for TcPKAR the yield of refolded protein (≤2 μM) was too low to carry out further experiments. Isothermal titration calorimetry (ITC) measurements showed high affinity binding of inosine and guanosine to both TbPKAR (**Figure 1d**) and LdPKAR1

(*Figure 1e*) with nanomolar $K_D$ values, matching closely the $EC_{50}$ values for kinase activation (*Table 1*). The close match of binding $K_D$ and activation $EC_{50}$ values for inosine and guanosine suggests that the $K_D$ measured for PKAR expressed in *E. coli* is an excellent proxy for the binding $K_D$ to the holo-enzyme. Adenosine, the weakest activator, is also the weakest binder. Inosine did not bind at all to human PKARIα, which bound cAMP with a $K_D$ of 23 nM (*Figure 1—figure supplement 1d*). LdPKAR1 did not bind to cAMP (*Figure 1—figure supplement 1d*) as shown before for TbPKAR (*Bachmaier et al., 2019*). The binding data thus support nucleoside-specificity and cAMP independence of the trypanosomatid PKAs. The stoichiometry of purine nucleoside binding to TbPKAR as calculated from ITC data was N ≈1, apparently lower than expected for the two binding sites occupied by inosine in the co-crystal structures (see below). We cannot exclude the possibility that a fraction of the refolded protein unfolds or aggregates after purification or is bound to remaining traces of the ligand and therefore not available for binding at the time of ITC analysis (see Materials and methods). The thermodynamic signature of nucleosides binding to kinetoplastid PKA resembles that of mammalian PKA bound by cAMP (*Figure 1—figure supplement 3*). The enthalpic contribution to binding (ΔH), indicating strong hydrogen bonding, is counteracted by a relatively large loss of entropy (TΔS), indicating bound-state conformational constrains.

## Structure of the nucleoside-binding pockets

To evaluate the binding mode, we solved the crystal structures of *T. cruzi* PKAR(200-503) and *T. brucei* PKAR(199-499) bound to inosine at 1.4 Å and 2.1 Å resolution, respectively (*Figure 2—figure supplement 1e, f,* ). Attempts to crystallize LdPKAR1(200–502) were unsuccessful. The structures of TbPKAR and TcPKAR show high overall similarity. Calculated RMSD of Cα alignment was 0.796 Å for the entire proteins and 0.281/0.342 Å for CNB-A and CNB-B, respectively. Residues that contribute to high affinity binding by interacting with the ribose moiety of inosine are identical in both structures (*Figure 2a and b*; *Figure 2—figure supplement 1g*; *Video 1*) and reside in a segment that we denoted ribose binding cassette (RBC), in analogy to the phosphate binding cassette (PBC) nomenclature for mammalian PKA (*Canaves and Taylor, 2002*). These residues in site A (308-320[TbPKAR], 309-321[TcPKAR]) and site B (432-445[TbPKAR], 433-446[TcPKAR]) engage in the same interactions with inosine as in our previously described co-crystal structure of TcPKAR(200-503) with 7-CN-7-C-Ino (PDB: 6FTF) (*Bachmaier et al., 2019*). Likewise, the 'capping' by π-stacking with the purine ring in both sites (Y371/483[TcPKAR], Y370/482[TbPKAR]) and the interacting residues in the lid-like αD helix are conserved. Thus, the binding mode of inosine and 7-CN-7C-Ino is almost identical. Minor differences are compatible with similar affinities of inosine and 7-CN-7-C-Ino. In site A, amino acid K294 donates a hydrogen bond to the cyano group of 7-CN-7-C-Ino (*Figure 2c*). When bound to inosine, however, a different side chain rotamer of K294 is preferred, and a hydrogen bond can now be formed with the keto group in position 6 of the purine ring (*Figure 2c*). In site B, the bulkiness of the cyano group displaces the side chain of Y485 by 1.5 Å, creating a small hydrophobic pocket able to fit C7-derivatives (*Bachmaier et al., 2019*; *Figure 2c*). Comparison of mammalian cAMP-bound PKARIα (PDB:1RGS) with nucleoside-bound kinetoplastid structures (PDB: 6FLO) clearly suggests that residues A202/R209[PKARIα] in site A and A326/R333[PKARIα] in site B are key to explain the altered ligand specificity of the kinetoplastid PKAR subunits (*Figure 2d*). The arginine residues 209/333[PKARIα] conserved in most PKARs are replaced by polar amino acids, and the alanine residues 202/326[PKARIα] are replaced by glutamates highly conserved in the kinetoplastid PKARs. The arginine in PKARIα neutralizes the negative charge of the phosphate in cAMP and also donates hydrogen bonds to the exocyclic oxygens of the cyclic phosphate. The glutamates in kinetoplastid RBCs interacts with the 3' and 5' OH groups of ribose. Moreover, the arginines 209/333[PKARIα] and the glutamates 311/435[TbPKAR] occupy the same spatial position in the structures (*Figure 2d*). A superposition of the mammalian and *T. brucei* structures shows a clash between the phosphate group of cAMP and the negatively charged side chain of E311/435[TbPKAR] (*Figure 2d* insets). The high-resolution crystal structures of *T. brucei* PKAR and *T. cruzi* PKAR thus provides a molecular rationale for absence of binding and activation by cAMP.

## Synthetic conversion of TbPKAR to cyclic nucleotide specificity

To identify the structural determinants of ligand specificity, we introduced site-directed amino acid changes in TbPKAR to restore binding and activation by cyclic nucleotides. Three residues in each of the binding sites were mutated: E311A, T318R, V319A in RBC-A and E435A, N442R, V443A in RBC-B

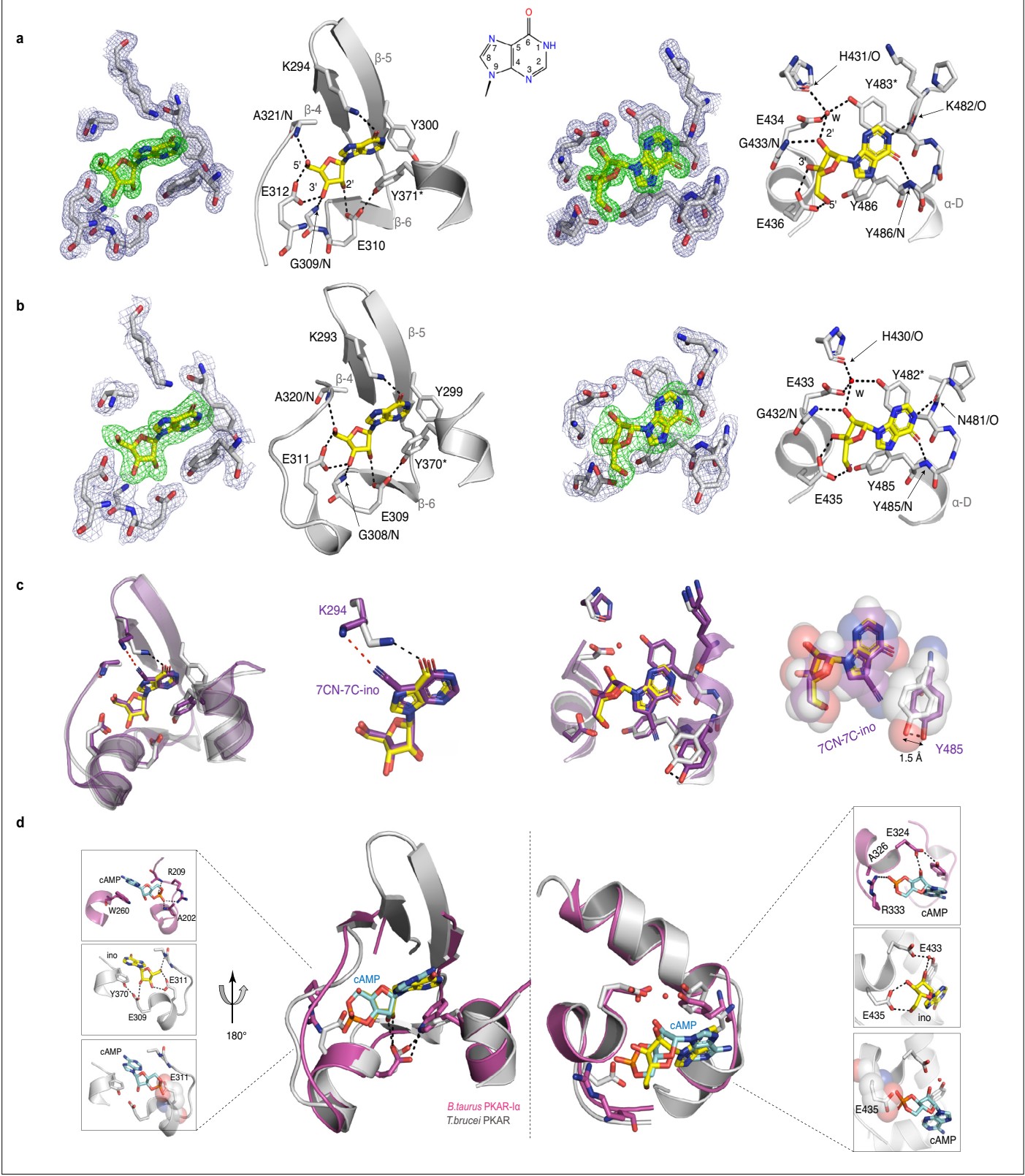

**Figure 2.** Crystal structures of *T. cruzi* and *T. brucei* PKAR bound to inosine. (**a**) Electron density (ED) maps of site A (left) and site B (right) of *T. cruzi* PKAR(200–503) and corresponding ball and stick models of the hydrogen bond network around the bound inosine molecule. The inosine molecule was modeled into the omit map (Fo-Fc, 3σ, green) in each binding site. The surrounding protein atoms are shown together with a 2Fo-Fc map (1σ, dark blue). The black dashed lines represent hydrogen bonds (≤3 Å cutoff). Residues G309; E310; E312 and A321 belong to Ribose Binding Cassette

*Figure 2 continued on next page*

Figure 2 continued

A (RBC-A), while G433; E434 and E436 are part of Ribose Binding Cassette B (RBC-B). Capping residues (Y371 and Y483) taking part in a π-stacking interaction with the hypoxanthine ring of inosine are marked with an asterisk. Purine ring nomenclature is shown in the middle. PDB: 6HYI. (**b**) *T. brucei* PKAR(199-499) displayed as in A, Residues G308, E309, E311, and A320 are part of RBC-A while G432, E433, and E435 belong to RBC-B. Capping residues (Y370 and Y482) are marked with an asterisk. PDB: 6FLO. (**c**) Structural alignment of inosine-bound *T. cruzi* PKAR (PDB: 6HYI; protein grey, inosine yellow) and 7-CN-7C-Ino-bound *T. cruzi* PKAR (PDB: 6FTF; protein and ligand in purple). The different ligand binding to K294 (A-site, left) and a 1.5 Å displacement of Y485 due to the bulky cyano group of 7-CN-7-C-Ino (B-site, right) are shown at two magnifications. (**d**) Structural alignment of TbPKAR (PDB: 6FLO; protein grey, inosine yellow) and *B. taurus* PKARIα (PDB: 1RGS; protein magenta, cAMP cyan) for binding sites A (left) and B (right). In the blow-up panels, ligand-protein interactions are highlighted for the mammalian PKARIα (upper panel), TbPKAR (middle panel), and TbPKAR overlayed with the cAMP ligand of the aligned PKARIα structure. A clash between the exocyclic oxygens of cAMP and the side chain of glutamate residues (faded sphere-representation) is seen in both binding sites.

The online version of this article includes the following source data and figure supplement(s) for figure 2:

**Figure supplement 1.** Protein purification and quality controls for ITC and crystallization experiments for TbPKAR, LdPKAR1, and TcPKAR.

**Figure supplement 1—source data 1.** Original Coomassie stained gel showing refolded (APO) LdPKAR1(200–502).

**Figure supplement 1—source data 2.** Original Coomassie stained gel showing refolded (APO) LdPKAR1(200–502) (labelled).

(mutant 1 in *Table 2*). In addition to the arginines and glutamates discussed above, a third position (V319/V443) that differs in kinetoplastid PKA compared to other eukaryotic PKAs (*Mohanty et al., 2015*; *Bachmaier et al., 2019*) was changed to alanine. The consensus PBC sequence of cAMP-dependent PKAs (*Canaves and Taylor, 2002*) was thereby restored. Mutant PKAR subunits were co-expressed with *T. brucei* catalytic subunits in *L. tarentolae*, and holoenzymes were tandem affinity purified. Kinase activation by nucleosides and cyclic nucleotides was measured (*Table 2*, *Figure 3—figure supplement 1*). Mutant 1 restored kinase activation by cIMP (EC$_{50}$ 340 nM) and reduced activation potency of inosine >21,000-fold compared to WT. Thus, we confirmed that replacing these key residues was sufficient for conversion to cyclic nucleotide specificity. Activation by cAMP was also restored, but at lower activation potency (EC$_{50}$ 33 µM). This corresponds to lower activation potency of adenosine compared to inosine for wild type TbPKA. The same ranking is also seen for the very low potencies of IMP, GMP, and AMP (*Tables 1 and 2*). To confirm the binding mode of cAMP to the converted binding site, mutant 6 of TbPKAR(199-499) carrying the triple replacements in site A was expressed in *E. coli* and co-crystallized with cAMP (*Figure 3—figure supplement 2a*). A molecule of cAMP was bound to site A and an inosine molecule (captured during expression in *E. coli*) to the unmodified site B (*Figure 3a*). Structural similarity of mutant 6 to wild type TbPKAR was very high (Cα RMSD = 0.430 Å). All hydrogen bonds to cAMP observed in the PKARIα structure (PDB: 1RGS, *Figure 3b*, right) were also present between cAMP and homologous residues in the A-site pocket of TbPKAR mutant 6 (*Figure 3b*, left and *Video 2*). The only remarkable difference is that cAMP binds in an *anti*-conformation in mutant 6 and in the *syn*-conformation in the mammalian PKAR. The E311A and V319A replacements created additional space inside the pocket to accommodate the bulky phosphate group of cAMP. An altered conformer of cysteine 278 and slight displacement of the loop between β–2 and β–3 in site-A allowed R318 to be accommodated so that it can interact with an exocyclic oxygen of cAMP (*Video 2*).

Simultaneous binding of cAMP and inosine to mutant 6 was supported by nanoDSF analysis. The refolded protein (APO form) had a low T$_m$ measured by nanoDSF, but T$_m$ raised by 14 °C upon addition of cAMP and by 28 °C upon addition of both cAMP and inosine (*Figure 3c*). cAMP also stabilized the refolded mutant 6 protein during purification (*Figure 3—figure supplement*

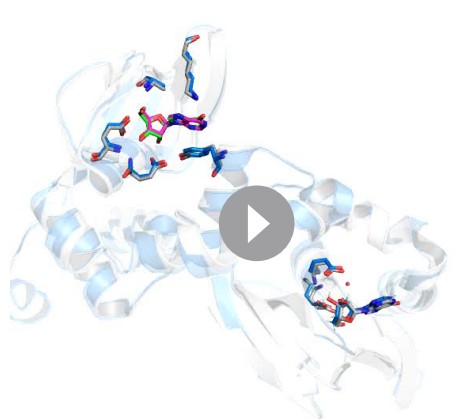

**Video 1.** Description: Alignment between TcPKAR (PDB: 6HYI, light blue) and TbPKAR (PDB:6FLO, chain B, light gray) displaying an RMSD of 0.909 Å calculated by PyMOL. Inosine is displayed in green and magenta in TbPKAR and TcPKAR, respectively.

https://elifesciences.org/articles/91040/figures#video1

**Table 2.** Activation of mutant TbPKA holoenzymes by different ligands.

| | TbPKA holoenzyme | $EC_{50}$ (95% CI) | | | | |
| | | Inosine | cIMP | Adenosine | Guanosine | cAMP |
|---|---|---|---|---|---|---|
| WT | RBC-A -GELELMYQTPTVA- RBC-B -GELEFLNNHANVA- | 18 (13–22) nM* | - ‡ | 5 (3-7) µM* | 0.14 (0.08–0.19) µM* | - ‡ |
| Mut ¶ | | | | | | |
| | RBC-A -GELALMYQTPRAA- RBC-B -GELAFLNNHARAA- | 300 (160–600) µM† | 0.34 (0.2–0.7) µM† | - ‡ | 370 (130–1000) µM† | 33 (26–45) µM† |
| | RBC-A -GELALMYQTPTVA- RBC-B -GELAFLNNHANVA- | 28 (23–35) µM† | nd** | nd | nd | - § |
| 3 | RBC-A -GELELMYQTPVVA- RBC-B -GELEFLNNHAVVA- | 28 (15–37) nM† | nd | nd | nd | nd |
| 4 | RBC-A -GELELMYQTPRVA- RBC-B -GELEFLNNHARVA- | 36 (29–44) nM† | 24 (22–26) µM† | nd | nd | - § |
| 5 | RBC-A -GELALMYQTPRVA- RBC-B -GELAFLNNHARVA- | 18 (15–20) µM† | 0.5 (0.3–0.6) µM† | nd | nd | 90 (71–115) µM† |
| 6 | RBC-A -GELALMYQTPRAA- RBC-B -GELEFLNNHANVA- | 0.2 (0.16–0.25) µM† | 14 (10–25) µM† | - ‡ | 1.2 (0.6–2.6) µM† | 25 (19–38) µM† |
| 7 | RBC-A -GELELMYQTPTVA- RBC-B -GELAFLNNHARAA- | 1.1 (0.9–1.4) µM† | 7 (5-11) µM† | 21 (13–30) µM† | 23 (17–32) µM† | - § |

*mean half activation constants ($EC_{50}$) and 95% confidence interval (95% CI) for ≥ 3 biological replicates.

†mean half activation constants ($EC_{50}$) and 95% confidence interval (95% CI) for technical triplicate of a single biological experiments.

‡no activation was detected up to a maximum concentration of 5 mM.

§no activation was detected up to a maximum concentration of 2 mM.

¶number given to the respective mutant; site-directed mutations indicated in red.

**not determined.

*2b*, c). Correct refolding of this mutant was indicated by identical $T_m$ after addition of cAMP plus inosine to native and refolded protein preparations and was confirmed by circular dichroism spectroscopy (*Figure 3—figure supplement 2d*). To evaluate the role of individual amino acids in the 'conversion set', single and double mutations were introduced at equivalent positions in RBC A and RBC B of TbPKAR and co-expressed with the *T. brucei* catalytic subunit PKAC1 in *L. tarentolae*. The tandem affinity purified holoenzymes were used for kinase assays to determine $EC_{50}$ values (*Table 2*, *Figure 3d*, *Figure 3—figure supplement 1*). Replacement of positions 318/442[TbPKAR] by arginines in both RBCs (mutant 4) was sufficient for response to cIMP ($EC_{50}$ 24 µM). To achieve cyclic nucleotide activation in the upper nM range the glutamates 311/435 needed to be replaced by alanines as well in mutant 5 (*Figure 3d*, *Table 2*). The potency of inosine was 1556-fold reduced by E311A/E435A[TbPKAR] (mutant 2) alone. In contrast, substituting the non-conserved amino acids at positions 318/442 by arginine (mutant 4) or a valine (mutant 3) did not have a significant effect on inosine response. The adjacent valine 319/443[TbPKAR] seems to contribute to activation by inosine 17-fold (compare mutants 1 and 5, *Table 2* and *Figure 3d*). In the B site this valine engages in hydrophobic interactions with the side chains of Y485 and K488, both belonging to the kinetoplastid-specific αD helix that supports inosine binding by sealing the binding pocket (*Video 3*). The $EC_{50}$ values of all mutants tested were almost 100-fold higher for cAMP than for cIMP, not surprising as inosine is a much better activator of the WT protein compared to adenosine (*Tables 1 and 2*).

## The αD helix is required for high affinity binding to the B-site

The binding and activation assays used so far average $K_D$ and $EC_{50}$ values over both the A-site and B-site of TbPKAR. As these sites are structurally not identical, we considered a kinetoplastid-specific feature of the B-site, the αD-helix (*Bachmaier et al., 2019*). This is a helical extension of αC beyond the small loop containing the capping residue Y482 at the end of αC that stacks with the purine ring of inosine (*Figure 4a*). In the ligand-bound structures of TbPKAR and TcPKAR, this helix docks to theβ-barrel of the B-site, covers the binding pocket and shields the ligand from solvent (*Figure 4a*,

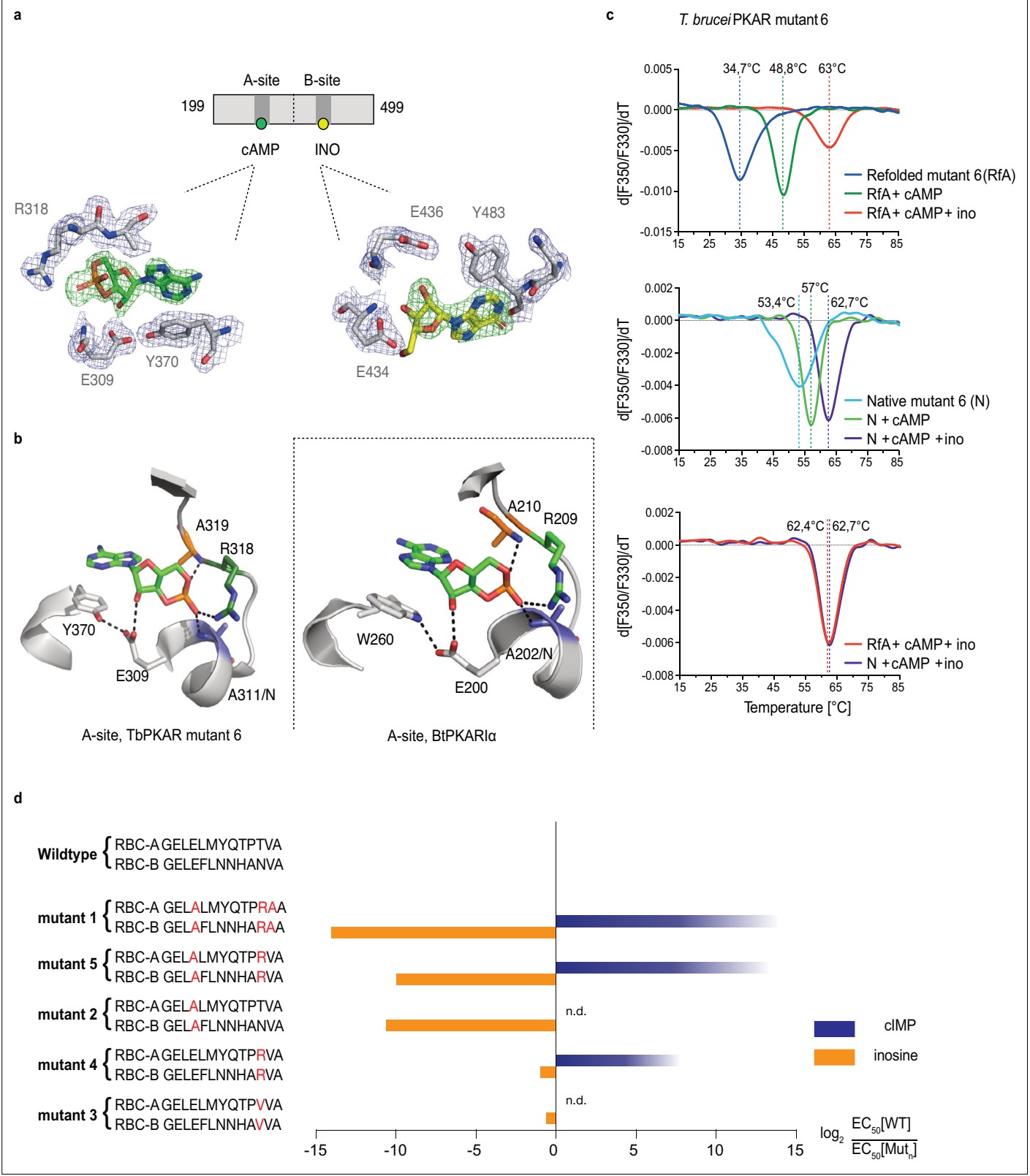

**Figure 3.** Conversion of TbPKAR to cyclic nucleotide specificity (**a**) Structure of ligand binding sites of TbPKAR(199-499) mutant 6 crystallized in presence of 1 mM cAMP (*Figure 3—figure supplement 2a*, PDB: 6H4G). The scheme above the electron density map highlights binding of cAMP to site A and inosine to site B. Below, the electron densities show the protein atoms inside the 2Fo-Fc (1σ, dark blue) map and ligands inside the Fo-Fc omit map (3σ, green). (**b**) Structural comparison between the A pocket of TbPKAR(199-499) mutant 6 (left) and BtPKARIα (right, PDB: 1RGS). The point

*Figure 3 continued on next page*

*Figure 3 continued*

mutations in mutant 6 are colored in purple (E311A), green (T318R) and orange (V319A). The same color code was used for the corresponding amino acids in BtPKARIα. Hydrogen bonds (3 Å) are indicated as dashed lines. (**c**) Thermal denaturation profiles (nanoDSF) of refolded APO (upper panel) and native mutant 6 TbPKAR(199-499) (middle panel) in the absence and presence of 1 mM ligands as indicated. The lower panel is a superposition of the thermal denaturation profiles of the two protein preparations (native and refolded APO) both incubated with 1 mM cAMP plus 1 mM inosine. (**d**) Mutational analysis of TbPKAR nucleoside binding sites. Relative kinase activation potency by inosine (orange) and cIMP (blue) is displayed as $\log_2$ of the $EC_{50}$[Wildtype]/$EC_{50}$ [Mutant$_n$] ratio on the x-axis. Since up to 5 mM cIMP did not activate the WT, this value was taken as minimal estimate of $EC_{50}$[WT] for cIMP. This uncertainty propagating into the calculated ratio is indicated by a color gradient at the right end of the columns. All data are taken from *Table 2*. Missing columns are not determined (n. d.). The sequences of RBC-A and RBC-B of mutants 1–5, with mutated amino acids highlighted in red, are shown on the left to the respective columns.

The online version of this article includes the following source data and figure supplement(s) for figure 3:

**Figure supplement 1.** Activation of mutant TbPKA holoenzymes by different ligands.

**Figure supplement 1—source data 1.** In vitro kinase activation assays of mutants 1–7 of *T. brucei* PKAR-PKAC1 holoenzyme by nucleosides and cyclic nucleotides.

**Figure supplement 2.** Protein purification and quality controls for ITC and crystallization experiments for TbPKAR mutant 6–8.

---

*Video 4*). Only one water molecule was found inside the binding pocket (*Figure 4a*). Two tyrosine residues (Y484, Y485) in αD are conserved in trypanosomatids, as are amino acids in the beta barrel of the B-site that are linked to these two amino acids via hydrogen bonds (*Figure 4b and c*). Y484 interacts with the backbone of V443 and R413 while Y485 forms two hydrogen bonds to the side chains of N438 and H440 (*Figure 4b*). In silico structure relaxation (*Figure 4d*) under an OPLS3 force field (Maestro-Schrödinger) showed two histidines (H440, H430) engaging in salt bridge interactions with the ribose-binding glutamates (E433 and E435), together forming a stable structure on which the αD-helix can dock (*Figure 4d*). The beta factor representation of TbPKAR suggests that proline 480 functions as a hinge between αC and αD, as the average displacement of P480 is higher than that of the other residues around it, likely correlating with higher mobility (*Figure 4e*). The αD-helix might therefore function as a lid to close the pocket and determine ligand affinity. This hypothesis was confirmed by ITC measurements of inosine binding to the Y484A/Y485A double mutant (mutant 8). To selectively measure nucleoside binding to the mutated Y484A/Y485A B-site, the A-site was made unavailable for nucleoside binding by using TbPKAR mutant 6 as context and refolding of the protein in the presence of cAMP (*Figure 3—figure supplement 2h*). Mutant 8 shows 82-fold decreased affinity for inosine (*Figure 4f*). The αD is therefore important for high binding affinity of 6-oxopurine nucleosides to the B-site.

## Site-selective binding and synergism of nucleosides

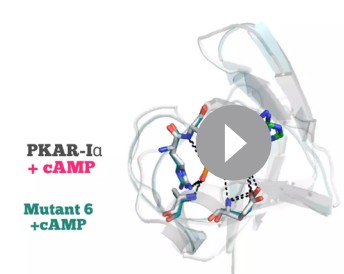

**Video 2.** Description: Alignment between A-site of PKARIα (PDB: 1RGS, gray, aa: 152–225) and A-site from mutant 6 (PDB: 6H4G, light green, aa: 259–332). In mammalian PKAR, cAMP binds in the *syn*-conformation, while in Mutant 6 it binds in the *anti*-conformation. Of particular note is Cys278 in TbPKAR mutant 6, which is significantly displaced to the newly inserted R318. Hydrogen bonds <3 Å are shown as black dashes.

https://elifesciences.org/articles/91040/figures#video2

The specific role of the αD-helix in site B prompted us to investigate binding affinities and ligand specificity of both sites individually. As interdomain CNB-A to CNB-B contacts are important for the allosteric activation mechanism of mammalian PKAR (*Akimoto et al., 2015*; *Berman et al., 2005*; *Kim et al., 2007*; *Malmstrom et al., 2015*), we analysed the contribution of each site in the R-subunit context. Mutant 6 with site A converted to cyclic nucleotide specificity (E311A, T318R, V319A) was blocked by excess of cAMP during refolding and was used to measure nucleoside binding to the non-mutated site B (*Figure 4g*). The corresponding mutant 7 with site B converted to cyclic nucleotide specificity was refolded in presence of cIMP to block this site and allow measurements of nucleoside binding affinity to the non-mutated A-site (*Figure 4h*). Correct refolding of mutant 6 and 7 was monitored by size exclusion chromatography and comparison of

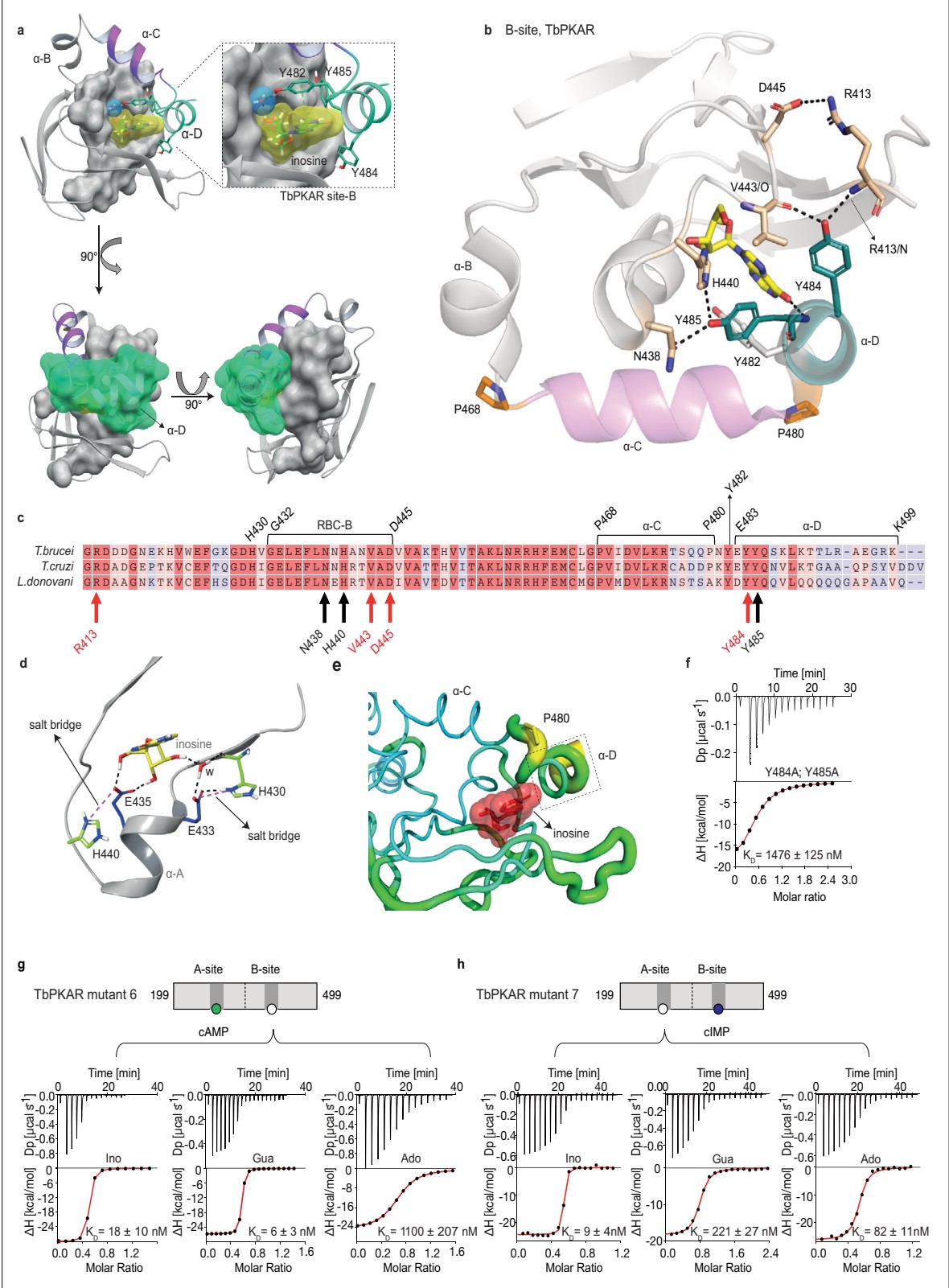

**Figure 4.** The αD helix of TbPKAR determines high binding affinity and ligand selectivity of the B-site. (**a**) C-terminal helices αB (grey), αC (purple) and αD (green) in CNB-B of TbPKAR, illustrating the lid-like position of the αD helix. The ribose binding cassette (RBC, residues 432–445) is shown in surface representation (grey). Inosine (yellow) and one water molecule (blue) are sandwiched between RBC-B and the αD helix (enlarged view in the blow-up panel). The 90° rotations of the structure show that the αC and αD helices are orthogonally positioned to each other. (**b**) The hydrogen-bond network

*Figure 4 continued on next page*

*Figure 4 continued*

formed between Y484 and Y485 in αD helix and amino acids in the beta barrel of site B. Hydrogen bonds (3 Å) are displayed as black dashed lines. (**c**) Sequence alignment of C-terminal domains from *T. brucei*, *T. cruzi*, and *L. donovani* PKARs. Only TbPKAR numbering is shown. Red arrows mark residues involved in the hydrogen bond network with Y484, black arrows mark residues involved in the hydrogen bond network with Y485. (**d**) Extended network of hydrogen bonds and salt bridges (purple dashed lines) between two conserved histidines (H430 and H440) and the ribose-binding glutamates E433 and E435. (**e**) Beta factor representation of TbPKAR site B showing a higher overall atom displacement in the crystal structure of the αD helix and in particular of P480. The Beta factor value increases from blue to red and from thin to thick, indicating an increase of atom displacement in the crystal. (**f**) Representative binding isotherm (ITC) for inosine binding to TbPKAR mutant 8 (mutant 6 with additional substitutions Y484A, Y485A). Data representation as in *Figure 1d*, the $K_D$ value is taken from . (**g**, **h**) Representative binding isotherms (ITC) for mutant 6 refolded in presence of 1 mM cAMP (**g**) and mutant 7 refolded in presence of 1 mM cIMP (**h**). Data representation as in *Figure 1d*, the $K_D$ values are taken from . For sequences of mutants see *Table 2*, for purity and non-aggregated state of R subunits see *Figure 3—figure supplement 2c*, e.

The online version of this article includes the following source data and figure supplement(s) for figure 4:

**Source data 1.** ITC measurements for inosine titrated to *T. brucei* PKAR mutant 8.

**Source data 2.** ITC measurements for inosine, guanosine, or adenosine titrated to *T. brucei* PKAR mutant 6.

**Source data 3.** ITC measurements for inosine, guanosine, or adenosine titrated to *T. brucei* PKAR mutant 7.

**Figure supplement 1.** Docking of nucleosides to A and B site of TbPKAR.

native and refolded protein by nanoDSF (*Figure 3c*; *Figure 3—figure supplement 2*). Inosine bound to mutant 6 and 7, and to the wildtype protein with similar high affinity ($K_D$ 9–18 nM). Adenosine had two orders of magnitude lower affinity for site B than inosine ($K_D$ 1.1 µM) but displayed high affinity for site A ($K_D$ 82 nM). In contrast, guanosine bound with highest affinity of all nucleosides to site B ($K_D$ 6 nM), but with 36-fold lower affinity to site A ($K_D$ 221 nM). The preference of mutant 7 for cIMP over cAMP also reflects the B-site specificity for 6-oxo purines. In silico docking of inosine, guanosine and adenosine to site A and B of TbPKAR (PDB: 6FLO, chain B) using GLIDE (*Friesner et al., 2004*) provided an explanation for the much lower binding affinity of adenosine to the B-site. Differences in interaction of the respective purine bases with the αD helix (*Figure 4—figure supplement 1*) include the hydrogen bonds of guanosine and inosine via the keto group at the C6 position to the backbone nitrogen of Y485. Since adenosine has an amino group in place of the C6 keto group, this specific interaction cannot take place. On the other hand, the C2 amino group of adenosine interacts with the keto group of N481 but apparently this interaction is not equivalent. Docking also suggests that the αD helix connects to guanosine, inosine and adenosine via three, two and one hydrogen bonds, respectively (*Figure 4—figure supplement 1*) which is perfectly compatible with weaker binding of adenosine to the B-site (*Figure 4g*). In the more solvent exposed site A, smaller differences in binding affinity of the three nucleosides (*Figure 4h*) correspond to smaller differences in the Glide G scores (*Figure 4—figure supplement 1*). In summary, molecular docking is compatible with the ITC data showing 37-fold binding selectivity of guanosine over adenosine at the B-site and 13-fold binding selectivity of adenosine over guanosine at the A-site (*Figure 4g and h*). An important implication of the site-selective binding of adenosine and guanosine is their possible synergism in kinase activation. This hypothesis was directly tested in kinase assays by determining the dose-response for adenosine in the presence of guanosine concentrations far below its $EC_{50}$. As seen in *Figure 5a*, the dose response curves were clearly left shifted (up to 20-fold) by guanosine addition. Thus, adenosine can activate TbPKA in the nanomolar range upon co-stimulation by very low concentrations of guanosine.

## Allosteric kinase activation

Comparing ligand binding data ($K_D$, , *Figure 1d*) and kinase activation $EC_{50}$ (*Table 2*, *Figure 1b*) of WT TbPKA we noticed that $K_D$ and $EC_{50}$ values matched very well for inosine and guanosine that bind with high affinity to the B-site, whereas a sixfold weaker activation compared to binding is seen for adenosine that preferentially binds to the A-site. This indicates that the B-site is the gate keeper and that our data are compatible with the model of allosteric regulation established for mammalian PKA (*Rehmann et al., 2007*; *Kim et al., 2007*), where a conformational change upon B-site binding gives access to the ligand at the A-site. We then calculated the $EC_{50}/K_D$ ratio (*Figure 5b*) for mutants 6 and 7 with either the B-site or the A-site intact. Both mutants show a high (>100) $EC_{50}/K_D$ ratio for nucleosides. This indicates that both binding domains are required in a ligand-bound conformation for efficient allosteric kinase activation by release of the catalytic subunit. The extreme (>4500) $EC_{50}/K_D$ ratio for adenosine and mutant 6 corresponds to the low affinity of adenosine to the B-site and confirms

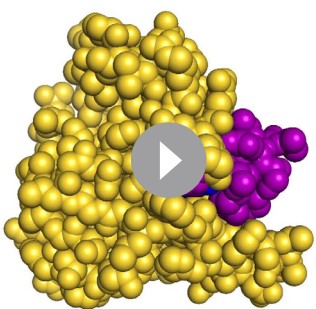

**Video 3.** Description: Sphere representation of the B-site from TbPKAR (PDB:6FLO, chain B, aa: 378–490) showing residues Y484, Y485, and K488 in the αD helix in purple, V443 in green, inosine in blue and the rest of the protein in yellow. V443 is sandwiched between the alpha-D helix and the beta barrel, taking part in hydrophobic contacts to both sides.
https://elifesciences.org/articles/91040/figures#video3

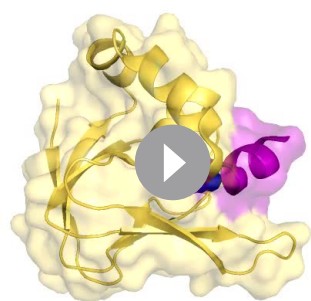

**Video 4.** Description: Surface representation of the B-site from TbPKAR (PDB:6FLO, chain B, aa: 378–490) showing an inosine molecule (blue spheres) locked inside the protein with no access to solvent. The αD helix (N481 to end) is depicted in dark purple.
https://elifesciences.org/articles/91040/figures#video4

the role of that site in initiating the conformation change. The $EC_{50}/K_D$ value of only 11 for mutant 6 and inosine is interpreted as weak binding of inosine to the mutated A binding site that we cannot exclude as the site is not blocked by cAMP in the kinase assays. Together, analysis of single binding site mutants and different nucleoside ligands in the context of the full length PKA provides strong support for conservation of an allosteric activation mechanism triggered by cooperative binding in hierarchical order, initiated by B-site binding.

## Ligands of trypanosome PKA in vivo

The biochemical and structural characterization of purified TbPKAR did not address the relative importance of the natural purine nucleosides for kinase activation in vivo in trypanosomes. The initial focus on inosine was due to the abundance of this nucleoside in *E. coli* and preloaded recombinant protein. We then quantified the loading of tagged PKAR with ligands upon rapid pulldown from *T. brucei* lysates. We expected at least a fraction of PKAR to be loaded with ligands due to the dynamic equilibrium between dissociated, ligand-bound and C-subunit-bound (holoenzyme complex) state. Tagged TbPKAR, but not endogenous TbPKAR was efficiently pulled down via the tag. Indeed, C subunits were co-purified, indicating only partial holoenzyme dissociation (*Figure 6—figure supplement 1*). Blood stream forms (BSF) and the procyclic fly vector stage of *T. brucei* (PCF) expressing tagged TbPKAR were used in these experiments and compared to matched isogenic wild type and *Δpkar/Δpkar* knock out controls. Nucleosides were released from PKAR bound to beads by boiling, then quantified by mass spectrometry using stable isotope labeled internal standards (*Figure 6*, *Figure 6—figure supplement 1*). In the procyclic stage (PCF), the relative amounts of nucleosides detected in the bead fraction were 63% adenosine, 30% guanosine, and 7% inosine (*Figure 6a*, *Figure 6—figure supplement 1a*). In the bloodstream stage (BSF) 94% guanosine, 6% inosine and only background level of adenosine were found (*Figure 6b*, *Figure 6—figure supplement 1b*). A priori the MS method did not exclude the additional presence of an unknown endogenous ligand of TbPKAR in trypanosomes. However, careful searches of the MS data sets for all known modified nucleosides detected in living systems from the MODOMICS database (*Boccaletto et al., 2018*) did not return significant hits absent in the blank. Thus, we propose that the nucleosides guanosine, adenosine and possibly inosine are endogenous ligands and likely activators of trypanosomatid PKA, probably acting synergistically.

## Discussion

PKA was the first protein kinase studied at the structural and mechanistic level and became a paradigm for allosteric kinase regulation by ligands (*Taylor et al., 2012*; *Taylor et al., 2021*). It is highly conserved through evolution, including its activation by cAMP and present in most species, except

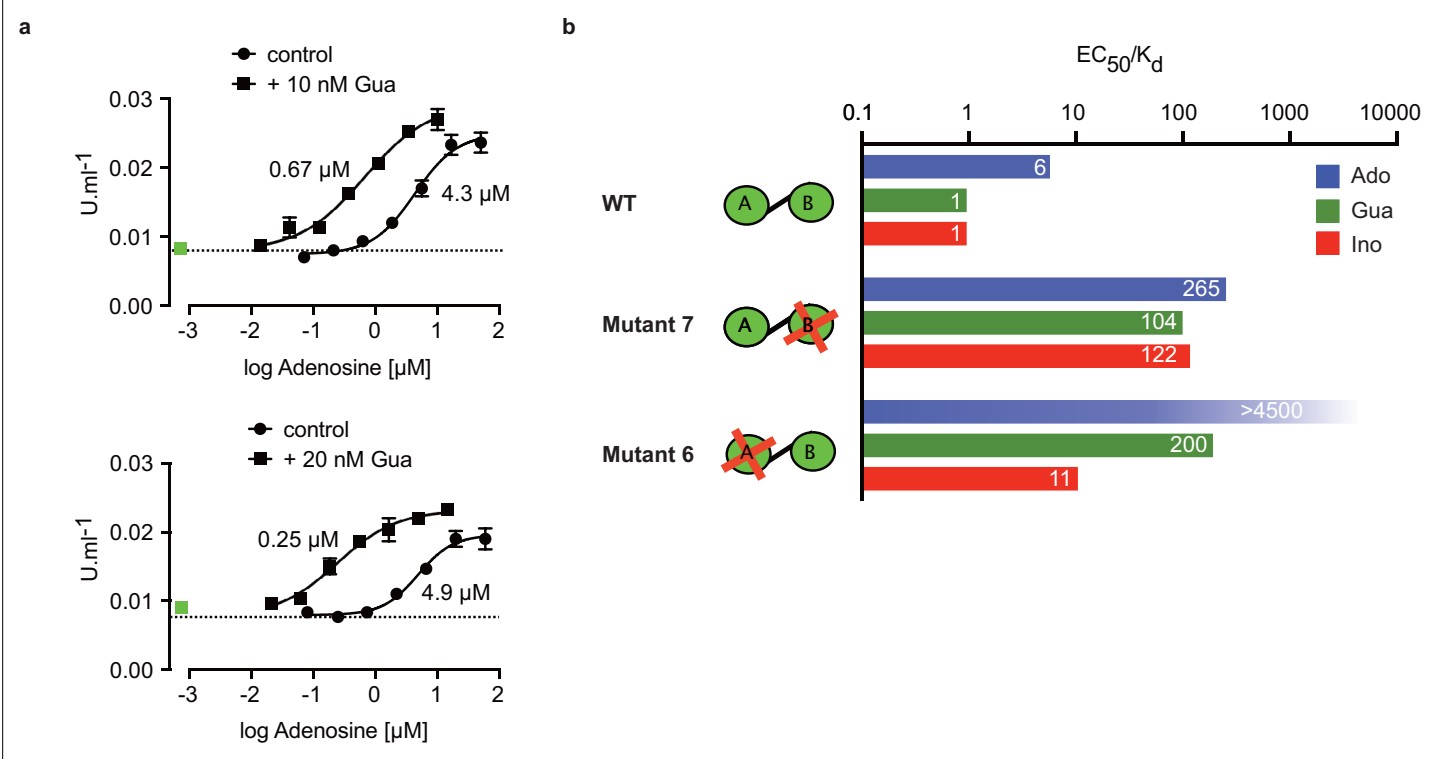

**Figure 5.** Binding site selectivity and synergism of nucleosides. (**a**) Dose-response curves for kinase activation of TbPKA by adenosine in presence of 10 nM or 20 nM guanosine. Error bars are m ± SD of technical triplicates, the calculated $EC_{50}$ values are given next to the respective curve. Basal kinase activity in the absence of any ligand is indicated by a horizontal dashed line. A green square (placed outside the log scale) represents the control with guanosine (10 or 20 nM) only. (**b**) Ratio of kinase activation over binding affinity ($EC_{50}/K_D$) for different purine nucleosides and individual binding sites A and B. Unavailable binding sites in mutants 6 and 7 are indicated by red crosses. Data are taken from *Table 2* and .

The online version of this article includes the following source data for figure 5:

**Source data 1.** In vitro kinase activation assays of *T. brucei* PKAR-PKAC1 holoenzyme by adenosine in presence of 10 nM or 20 nM guanosine.

plants. Here, we show that in the phylogenetically distant protozoan group *Trypanosomatida* nucleosides have replaced cyclic nucleotides as ligands of PKA. Inosine, guanosine and adenosine bind with high affinity to the regulatory subunit PKAR and efficiently activate PKA of *T. brucei*, *T. cruzi*, and *Leishmania spp*. Site-selective binding affinities and synergism of guanosine and adenosine suggest a new second messenger signaling pathway or nucleoside sensing mechanism in *Trypanosomatida*.

## The CNB domain - a versatile ligand binding domain

In trypanosomatid PKAR orthologs, few residues in each CNB domain systematically deviate from the consensus of the cyclic nucleotide binding motif (*Mohanty et al., 2015*; *Canaves and Taylor, 2002*). By mutagenesis of these residues (E311, T318, V319 in RBC A and E435, N442, V443 in RBC B of TbPKA), we were able to restore binding and kinase activation by cyclic nucleotides and structurally interpret the determinants of altered ligand specificity. Whereas the crystal structures predict that the most critical glutamates E311[RBC:A] and E435[RBC:B] required for nucleoside binding preference would clash with the cyclic phosphate of cAMP (*Figure 2d*), kinase activation at high concentration suggests that this incompatibility is not absolute (mutant 4, *Figure 3d*), as expected for a dynamic structure in vivo. CNBs are present in most species and in a broad variety of proteins, reaching from protein kinases (*Diller et al., 2001*; *Su et al., 1995*) to ion channels (*Zagotta et al., 2003*) and transcription factors, such as the catabolite activator protein (CAP) in bacteria (*Kannan et al., 2007*; *Passner and Steitz, 1997*). The ancient CNB fold has been described as core module for allosteric regulation by cyclic nucleotides (*Berman et al., 2005*; *Kannan et al., 2007*). Here, we propose that the CNB is a module for allosteric regulation by a broader spectrum of small ligands. This is reminiscent of other families of ligand binding proteins like the G-protein-coupled receptors or steroid hormone binding

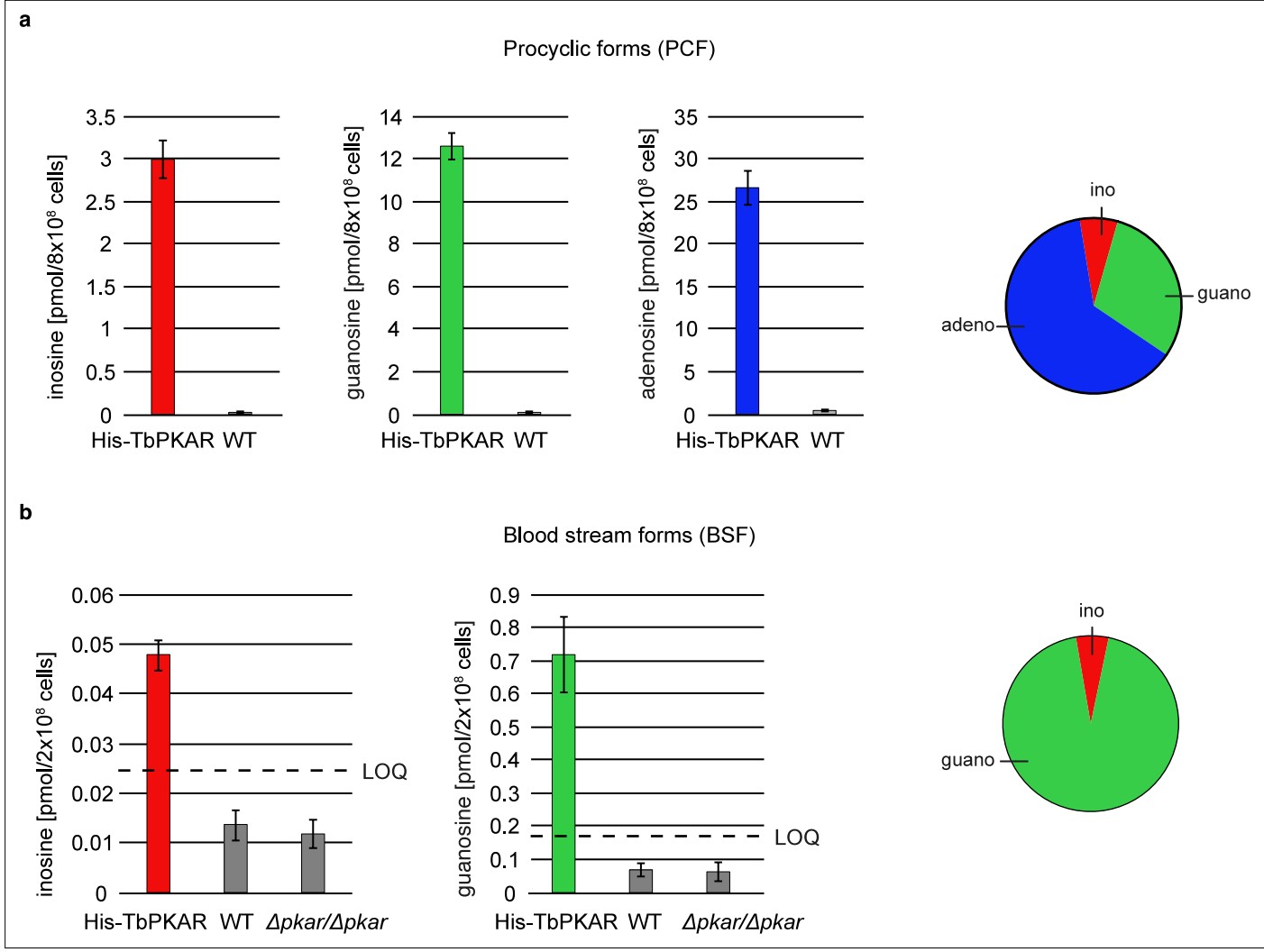

**Figure 6.** Quantification of ligands bound to TbPKAR in lysed cells HPLC-MS-based quantification of nucleoside amounts released from boiled His-tagged TbPKAR pulled down from lysed *T. brucei* (see *Figure 6—figure supplement 1*). Inosine (red), guanosine (green), and adenosine (blue) were quantified using stable isotope-labeled internal standards. Error bars indicate SD from three biological replicates. Note the different Y-axis scales. Pie charts on the right show the relative amounts of nucleosides detected and quantified. (**a**) Procyclic stage *T. brucei* strain EATRO1125 expressing His-TbPKAR and parental control cells. Pulled down nucleosides from the control cell line were in the range of water blanks. (**b**) Bloodstream stage *T. brucei* MITat 1.2 single marker line expressing His-TbPKAR, parental control cells and isogenic *Δtbpkar/Δtbpkar* cells devoid of endogenous PKAR. The limit of quantification (LOQ), defined by the linear part of the standard curves for stable isotope-labeled nucleoside references, is given by a dashed line. Adenosine was below the LOQ.

The online version of this article includes the following source data and figure supplement(s) for figure 6:

**Figure supplement 1.** HPLC-MS quantification of ligands bound to TbPKAR in parasite lysates.

**Figure supplement 1—source data 1.** Original file for the western blot analysis in *Figure 6—figure supplement 1*.

**Figure supplement 1—source data 2.** Original file for the western blot analysis in *Figure 6—figure supplement 1* with highlighted bands and sample labels.

domains in transcription factors that have been initially characterized by a limited set of ligands. Diverse ligands have later been identified for the 'orphan' members of those families (*Davenport et al., 2013*). Future investigations might identify additional CNB domain ligands also outside the *Trypanosomatida* group.

In contrast to residues important for ligand specificity, the π-stacking interaction by the so-called capping residues (*Wu et al., 2004*; *Kim et al., 2007*) with the purine ring of cAMP or nucleosides are well conserved between mammalian and trypanosomatids. Single mutations of the mammalian PKA

capping residues W260 [RIα:CNB-A] or Y371[RIα:CNB-B] that interact with cAMP reduced activation potency by 4.6 and ninefold, respectively and influence the cooperativity of the two binding sites (*Kim et al., 2007*). The importance of stacking interactions with Y371 in site A and Y482 in site B of TbPKAR explains why allopurinol riboside, a purine derivative very similar to inosine with reduced delocalized electron system, is 120-fold less potent than inosine (*Table 1*).

The biochemical and structural evidence for a distinct ligand specificity of trypanosomatid PKA that is provided here, will resolve a very controversial issue: whereas difficulties to detect cAMP-dependent kinase activity in *T. brucei* and *Leishmania* were reported long ago (*Walter, 1978*; *Banerjee and Sarkar, 2001*), cAMP regulation of *T. cruzi* and *Leishmania* PKA activity and binding of cAMP to LdPKAR1 in the µM range have been proposed by others (*Huang et al., 2006*; *Bhattacharya et al., 2012*). Our data contradict the latter reports and possible technical reasons have been discussed previously (*Bachmaier et al., 2019*; *Bachmaier and Boshart, 2013*).

## The tail makes the affinity difference

The very high affinity of nucleosides to the B-site was surprising as the ionic interaction of the cyclic phosphate deep in the pocket is important for strong binding of cAMP in mammalian PKA (*Su et al., 1995*; *Herberg et al., 1996*). The C-terminal extension (αD helix) is so far only found in trypanosomatid PKAR and contains the conserved sequence (K/N)YxYY. Our crystal structures show that this helix covers the B binding site in the ligand-bound state like a lid and shields the ligand from solvent (*Video 4*). Inside the binding pocket, the capping residue (Y482) π-stacks with the purine ring that may additionally engage in T-shape π-stacking interaction with Y484 and Y485 from the αD helix (*Figure 4b*). C-terminal extensions are found in some PKAR subunits for example RIIß (PDB: 1CX4; *Diller et al., 2001*) or *Plasmodium falciparum* PKAR (PDB: 5K8S; *Littler et al., 2016*). These differ from the αD helix in that they just seem to prolong the αC helix. In contrast, the αD helix is separated from αC by a proline resulting in a kink that positions the αD helix on top of the binding pocket. The helix is attached to the binding pocket by several hydrogen bonds donated by the conserved tyrosines Y484 and Y485 (*Figure 4b*). Consequently, the Y484A/Y485A mutation reduced the binding affinity for inosine to site B drastically. The αD helix lid mechanism may compensate for the weaker bonds of the ribose moiety of nucleosides deep in the pocket, whereas cAMP binding is stabilized by a strong ionic interaction with the phosphate in cAMP-dependent PKARs.

## Site selectivity and synergism of nucleosides

The binding data for trypanosomatid PKARs show a clear affinity ranking of the three purine nucleosides: $K_D$(ino)<$K_D$(guano)<$K_D$(adeno). The preference of inosine and guanosine over adenosine is most striking for the B-site. The structures and molecular docking show that the αD-helix contributes to the high affinity of guanosine and inosine, as in contrast to the adenine base, the 6-oxopurines can form several hydrogen bonds to the αD-helix. This preference is also seen as a 97-fold difference in kinase activation between cIMP and cAMP in mutant 1 (*Table 2*). Initially, binding of nucleosides to TbPKAR and LdPKAR1 was measured with recombinant proteins containing the complete C-terminus with both CNB domains. This averages over the affinity of two binding sites. To determine single binding site affinities, previous work on mammalian PKA used individual expressed CNB domains (*Moll et al., 2007*; *Lorenz et al., 2017b*). Here, we blocked either the A (mutant 6) or B site (mutant 7) of TbPKAR by conversion to cyclic nucleotide specificity to measure binding to the other site in the context of the intact protein. This strategy reduces the risk of protein truncation artefacts, but we are aware of the limitation of measuring binding to a 'primed' protein (the other binding site is occupied). The true affinity to the B-site in the holoenzyme (APO) state available in vivo cannot be easily determined. This in mind, a strong site preference of adenosine for site A and guanosine for site B was observed, whereas inosine bound equally to both sites. The αD-helix seems important for guanosine preference in site B (see above), but single amino acids contributing to adenosine preference in site A could not be identified by in-silico docking experiments (*Figure 4—figure supplement 1*). A co-crystal structure of TbPKAR with adenosine is not yet available. The site selective binding of adenosine and guanosine to sites A and B, respectively, of *T. brucei* PKAR is reminiscent of site selectivity of cyclic nucleotide analogues for mammalian PKAR. A synergistic effect on kinase activation of these synthetic compounds was exploited for development of potential anti-proliferative drugs (*Schwede et al., 2000*; *Cho-Chung et al., 1989*; *Huseby et al., 2011*; *Gausdal et al., 2013*). Priming mammalian RI

and RII isoenzymes with B-site selective cyclic nucleotides led to an increase of activation potency of A-site selective compounds (*Ogreid et al., 1989*; *Dostmann et al., 1990*; *Corbin et al., 1986*). We observed a comparable synergistic effect upon priming of TbPKA with 10–20 nM of the B-site selective guanosine, resulting in a 6–20-fold shift in activation potency by the A-site selective adenosine. In contrast to pharmacological synergism of drugs acting on mammalian PKA, the synergism of two endogenous ligands present in trypanosomes may have in vivo relevance by providing a logical AND switch to respond to and integrate over the two most important purine nucleosides in the cell.

## Allosteric regulation of PKA

A detailed model for the allosteric activation mechanism of mammalian PKA has been elaborated over many years (*Su et al., 1995*; *Kim et al., 2007*; *Taylor et al., 2021*; *Rehmann et al., 2007*). Is this activation mode also applicable to the nucleoside dependent PKA of trypanosomatids? The crystal structures of inosine bound TbPKAR and mammalian PKARIα (PDB: 1RGS) are highly homologous (rmds = 3.2 Å). Key sequence features implicated in the allosteric regulation, such as the salt bridge (E371, R475$^{TbPKAR}$) keeping the B/C helix extended in the apo conformation, the capping residues (Y370 and Y482) that participate in ligand binding by π stacking (*Kim et al., 2007*; *Wu et al., 2004*) as well as residues involved in R-C interaction for example the inhibitory sequence (RRTTV, res. 201–206 in TbPKAR) (*Kim et al., 2005*; *Kim et al., 2007*) are conserved. The binding and kinase activation data for single site mutants of TbPKA (*Figure 5b*) clearly show that nucleoside binding to both sites is required for efficient kinase activation and suggests that the B-site has a 'gatekeeper' function that initiates the conformational change and leads to accessibility of the A-site, like in mammalian PKA. Subsequent ligand binding to the A-site then triggers dissociation of the catalytic subunit (*Herberg et al., 1996*; *Kim et al., 2005*; *Kim et al., 2007*) and thereby releases the kinase from (auto)inhibition. The results of our single site mutant analysis of TbPKA (*Figure 5b*) are perfectly compatible with this model. Therefore, the basic allosteric mechanism seems to be conserved and may predate in evolution the separation of PKA into different CNB ligand specificities. More insight into the conformational detail of kinase activation by nucleosides will require the structure of a trypanosomatid PKA holoenzyme complex.

## Nucleosides as second messengers?

Inosine, guanosine, and adenosine were shown to bind to TbPKAR in vivo (*Figure 6*), whereas no compounds from an exhaustive list of nucleoside analogues previously identified in living organisms (MODOMICS database; *Dunin-Horkawicz et al., 2006*; *Boccaletto et al., 2022*; *Boccaletto et al., 2018*) did match MS spectra from pulled down bound material. We may have missed a low affinity or labile ligand not captured by the pulldown procedure. However, in favour of a second messenger role of nucleosides, their binding affinities and activation potencies match well the affinity and potency of cAMP for mammalian PKA (*Figure 1*, *Table 1*). Furthermore, an unidentified second messenger would compete with the nucleosides in the cell and therefore require even higher affinity. Obviously, PKA might also serve as an intracellular nucleoside gauge. Trypanosomes are purine auxotroph and require sensitive regulation of purine uptake (*Rico-Jiménez et al., 2021*). Reconciling a classical second messenger function of nucleosides with their role of metabolites in the cell seems less of a conceptual problem. First, an increasing number of metabolites with signaling function emerge (*Baker and Rutter, 2023*) and second, subcellular compartmentalization of signal generation/degradation is well established for cAMP signaling and PKA that act mostly in microdomains (*Buxton and Brunton, 1983*; *Zaccolo et al., 2021*; *Paolocci and Zaccolo, 2023*; *Musheshe et al., 2018*). Compartmentalization of cAMP signaling also characterizes the trypanosome flagellum that serves as sensory organelle (*Bachmaier et al., 2022*; *Ooi and Bastin, 2013*; *Oberholzer et al., 2010*; *Shaw et al., 2022*). Microdomains enriched in nucleoside salvage pathway enzymes may facilitate kinase activation or signal termination by rapidly generating or degrading specific nucleosides. The compartmentalized nucleotide metabolism (*Ginger et al., 2008*) in the flagellum exemplifies such a scenario. In fact, PKA of the trypanosomatid species analysed here is predominantly localized in the flagellum (*Bachmaier et al., 2016*; *Oberholzer et al., 2011*; *Billington et al., 2023*), The subcellular distribution of nucleosides in trypanosomes is not known and even reliable values for total guanosine and inosine per cell are unavailable in the literature. Only adenosine was measured in the bloodstream stage of *T. brucei* in the range of 12–28 µM depending on the growth medium (*Kim et al., 2015*). Taking the binding affinities

into account, we can use the relative amounts of bound nucleosides pulled down with TbPKAR from cell lysates (*Figure 6*) as proxy for relative nucleoside concentrations in PKA containing compartments. Only a small fraction of PKAR bound inosine in vivo in both life cycle stages, indicating that this high affinity binder is not readily available for PKAR in the parasites. This is in contrast to *E. coli* where inosine seems more abundant (extrapolated from *Bennett et al., 2009*) resulting in inosine bound to trypanosomatid PKAR when expressed in this heterologous system. In *T. brucei* bloodstream forms guanosine was captured but adenosine remained below the level of detection (*Figure 6*). In procyclic forms, guanosine and adenosine were both captured by PKAR, indicating a much higher adenosine/guanosine ratio in this life cycle stage. Based on these estimations, guanosine seems to be the primary physiological ligand in BSF, whereas adenosine and guanosine likely synergize to activate PKA in PCF (*Figure 5a*). Most PKAR pulled down from PCF would then have adenosine in the A-site and guanosine in the B-site. Termination of signaling may be supported by PKA subunit turnover since rapid degradation of the TbPKAC1 was seen upon depletion of TbPKAR by RNAi (*Bachmaier et al., 2019*).

As alternatives to a classical second messenger role of nucleosides, physiological activation of the PKA holoenzyme in vivo may be co-activated by nucleoside binding together with a second trigger like a posttranslational modification, redox state, specific protein-protein interaction or kinase regulation by liquid–liquid phase separation (*López-Palacios and Andersen, 2023*; *Hardy et al., 2023*). These triggers may allosterically shift the affinity or may be required for the final activating conformational change upon binding (*Khamina et al., 2022*), giving nucleosides a more auxiliary role in the allosteric kinase regulation. In trypanosomatids abundant and regulated phosphorylation of PKA subunits has been reported (*Tsigankov et al., 2014*; *Urbaniak et al., 2013*). Kinase regulation may then even not require a signal-related change of the intracellular nucleoside concentration. The reasoning is inspired by the role of specific phosphorylations of mammalian PKA (*Haushalter et al., 2018*) and the phenomenon of allosteric pluripotency described for the analogue Rp-cAMP, that acts as an inhibitor or activator depending on MgATP concentrations (*Dostmann and Taylor, 1991*; *Byun et al., 2020a*). Mechanistically, the opposite effects of Rp-cAMP are explained by formation of energetically stabilized mixed intermediate states of the kinase, in which CNB-A and CNB-B adopt different conformational states (*Byun et al., 2020b*; *Akimoto et al., 2015*). Similar intermediate states were also reported for *Plasmodium falciparum* PKG (*Byun et al., 2020c*). Models for activation of the kinetoplastid PKA remain speculative as long as the upstream signaling that leads to PKA activation in vivo has not been elucidated in these organism. Future research will use kinase activation as readout for genome-wide screening to detect upstream pathway components regulating the nucleoside-dependent PKA.

# Materials and methods

**Key resources table**

| Reagent type (species) or resource | Designation | Source or reference | Identifiers | Additional information |
|---|---|---|---|---|
| Gene (*T. brucei*) | TbPKAR | TriTrypDB | Tb927.11.4610 | |
| Gene (*T. brucei*) | TbPKAC1 | TriTrypDB | Tb927.9.11100 | |
| Gene (*T. cruzi*) | TcPKAR1 | TriTrypDB | TcCLB.506227.150 | |
| Gene (*T. cruzi*) | TcPKAC2 | TriTrypDB | TcCLB.508461.280 | |
| Gene (*L. donovani*) | LdPKAR1 | TriTrypDB | LdBPK_130160.1 | |
| Gene (*L. donovani*) | LdPKAC1 | TriTrypDB | LINF_350045600 | |
| strain, strain background (*E. coli*) | Rosetta (DE3) | Novagen | SKU: 70954–3 | Electrocompetent cells |
| Cell line (*L. tarentolae*) | LEXSY T7-TR | Jena Bioscience | Cat.No.: LT-110 | |

*Continued on next page*

*Continued*

| Reagent type (species) or resource | Designation | Source or reference | Identifiers | Additional information |
|---|---|---|---|---|
| Cell line (*T. brucei*) | *T. brucei brucei* stock Lister 427 clone MiTat 1.2 | 10.1017/S0031182000046540 | | All *T. brucei brucei* cell lines are derived in the laboratory from *T. brucei brucei* stock Lister 427 clone MiTat 1.2, originally obtained from G. Cross, NY |
| Cell line (*T. brucei*) | *T. brucei* EATRO 1125 AnTat1.1 90:13 | 10.1101/gad.323404 | | |
| Cell line (*T. brucei*) | MITat1.2SM 6HIS TbPKAR | This paper | | MITat1.2_SM blood stream forms (BSF) of *T. brucei* with expression of TbPKAR fused to a 6xHis tag |
| Cell line (*T. brucei*) | EATRO11252T7 6HIS TbPKAR | This paper | | EATRO11252T7 procyclic forms (PCF) of *T. brucei* with expression of TbPKAR fused to a 6xHis tag |
| Cell line (*T. brucei*) | MITat1.2SM PKAR-KO | 10.1038/s41467-019-09338-z | | MITat1.2_SM BSF PKAR knock out |
| Transfected construct (*T. brucei*) | pLEW100v5b1d-BSD_6His-Tev TbPKAR | This paper | | Construct cloned and transfected into MITat1.2SM (BSF) |
| Transfected construct (*T. brucei*) | pHD1146-puro_6xHis-Tev TbPKAR | This paper | | Construct cloned and transfected into EATRO11252T7 (PCF) |
| Antibody | anti-PKAR (rabbit polyclonal) | 10.1038/s41467-019-09338-z | | 1:500 |
| Antibody | Anti-PKAC1/2 (rabbit polyclonal) | 10.1038/s41467-019-09338-z | | 1:500-1:1000 |
| Antibody | Anti-6x His tag (mouse monoclonal) | Thermofisher scientific | | 1:1000 |
| Antibody | IRDye 800CW anti-mouse IgG (goat polyclonal) | LICOR | Cat# 925–32210 | 1:5000 |
| Antibody | IRDye 680LT anti-rabbit IgG (goat polyclonal) | LICOR | Cat# 925–69021 | 1:5000 |
| Antibody | Alexa Fluor 680 anti-rabbit IgG (goat Superclonal Recombinant) | ThermoFisher | Cat# A27042 | 1:5000 |
| Recombinant DNA reagent | pLEXSY_I-ble3 (vector) | Jena Bioscience | Cat# EGE-244 | |
| Recombinant DNA reagent | pLEXY_I-neo3 (vector) | Jena Bioscience | Cat# EGE-245 | |
| Recombinant DNA reagent | pLEXSY_I-ble3_TbPKAR (plasmid) | 10.1038/s41467-019-09338-z | | Transfected in LEXSY expression system |
| Recombinant DNA reagent | pLEXSY_I-ble3_TbPKAR_mutant 1–7 (plasmids) | This Paper | | Transfected in LEXSY expression system; for specific mutation inserted see *Table 2* |
| Recombinant DNA reagent | pLEXSY_I-ble3_TcPKAR1 (plasmid) | This paper | | Transfected in LEXSY expression system |
| Recombinant DNA reagent | pLEXSY_I-ble3_LdPKAR1 (plasmid) | This paper | | Transfected in LEXSY expression system |
| Recombinant DNA reagent | pLEXY_I-neo3_TbPKAC (plasmid) | 10.1038/s41467-019-09338-z | | Transfected in LEXSY expression system |
| Recombinant DNA reagent | pLEXY_I-neo3_TcPKAC2 (plasmid) | This paper | | Transfected in LEXSY expression system |
| Recombinant DNA reagent | pLEXY_I-neo3_LdPKAC1 (plasmid) | This paper | | Transfected in LEXSY expression system |
| Recombinant DNA reagent | pETDuet-1 DNA-Novagen (vector) | Sigma-Aldrich Novagen | SKU 71146 | |

*Continued on next page*

*Continued*

| Reagent type (species) or resource | Designation | Source or reference | Identifiers | Additional information |
|---|---|---|---|---|
| Recombinant DNA reagent | pET_SUMO Expression system | ThermoFisher Scientific | K30001 | |
| Recombinant DNA reagent | pETDuet-1_TbPKAR(199-499) | 10.1038/s41467-019-09338-z | | Recombinant expression of TbPKAR(199-499) in *E. coli* |
| Recombinant DNA reagent | pETDuet-1_TbPKAR(199-499)_mutant 6 | This paper | | Recombinant expression of TbPKAR(199-499) mutant 6 in *E. coli* |
| Recombinant DNA reagent | pETDuet-1_TbPKAR(199-499)_mutant 7 | This paper | | Recombinant expression of TbPKAR(199-499) in *E. coli* |
| Recombinant DNA reagent | pETDuet-1_TbPKAR(199-499)_mutant 8 | This paper | | Recombinant expression of TbPKAR(199-499) in *E. coli* |
| Recombinant DNA reagent | pETDuet-1_TcPKAR1(200–503) | 10.1038/s41467-019-09338-z | | Recombinant expression of TcPKAR(200–5003) in *E. coli* |
| Recombinant DNA reagent | pETDuet-1_HsPKAR1α | PMID:8393867 | | Recombinant expression of HsPKAR1α in *E. coli* |
| Recombinant DNA reagent | pET-11_Sumo3_LdPKAR1(200–502) | This paper | | Recombinant expression of LdPKAR1(200–502) in *E. coli* |
| Peptide, recombinant protein | TEV Protease | NEB | Cat# 8112 S | Cleavage of HIS-Tag in recombinant protein |
| Peptide, recombinant protein | Sumo Protease | Sigma-Aldrich | SKUSAE0067-2500UN | Cleavage of SUMO Tag in recombinant protein |
| Commercial assay or kit | Hi Yield Plasmid Mini DNA Isolationkit | Süd-Laborbedarf GmBH, Germany | Art-Nr.: 30 HYPD100 | |
| Chemical compound, drug | Inosine | Sigma Aldrich | Cat# 200-390-4 | |
| Chemical compound, drug | Guanosine | Sigma Aldrich | Cat# 204-227-8 | |
| Chemical compound, drug | Adenosine | Sigma Aldrich | Cat# 93029 | |
| Chemical compound, drug | cAMP | Biolog Life Science Institute | Cat# A 001 H | |
| Chemical compound, drug | cGMP | Biolog Life Science Institute | Cat# G 001 | |
| Chemical compound, drug | cIMP | Biolog Life Science Institute | Cat# I 001 | |
| Chemical compound, drug | AMP | Sigma Aldrich | Cat# 54612 | |
| Chemical compound, drug | GMP | Santa Cruz Biotechnology | Cat# 226-914-1 | |
| Chemical compound, drug | IMP | Sigma Aldrich | Cat# 57510 | |
| Chemical compound, drug | 2'-deoxyadenosine | Sigma Aldrich | Cat# D7400-250MG | |
| Chemical compound, drug | 3'-deoxyadenosine | Sigma Aldrich | Cat# C3394 | |
| Chemical compound, drug | 5'-deoxyadenosine | Sigma Aldrich | D1771 | |

*Continued on next page*

*Continued*

| Reagent type (species) or resource | Designation | Source or reference | Identifiers | Additional information |
|---|---|---|---|---|
| Chemical compound, drug | Nebularine | Santa Cruz Biotechnology | Cat# sc-208087 | |
| Chemical compound, drug | Allopurinol riboside | Santa Cruz Biotechnology | Cat# sc-217610 | |
| Chemical compound, drug | Xanthosine | Sigma Aldrich | Cat# X0750 | |
| Chemical compound, drug | Uridin | Sigma Aldrich | Cat# U3750 | |
| Chemical compound, drug | Cytidin | Sigma Aldrich | Cat# C122106 | |
| Chemical compound, drug | 13C5-labeled Inosin | Omicron Biochemicals Inc | | 10.1038/s41596-018-0094-6 |
| Chemical compound, drug | 13C5-labeled Guanosin | Omicron Biochemicals Inc | | 10.1038/s41596-018-0094-6 |
| Chemical compound, drug | 13C5-labeled Adenosin | Omicron Biochemicals Inc | | 10.1038/s41596-018-0094-6 |
| Software, algorithm | GraphPad Prism 7.0 | GraphPad | | Statistical testing |
| Software, algorithm | Phenix | 10.1107/S0907444909052925 | | Model refinement |
| Software, algorithm | Coot | 10.1107/S0907444909052925 | | Manual model building of protein structure |
| Software, algorithm | Glide (Maestro) | Schroedinger LLC, New York, NY, 2023 | | Molecular docking |
| Software, algorithm | The PyMOL Molecular Graphics System (Version 2.0) | Schrödinger, LLC | | Illustration of structural figures |

## Preparation of PKA holoenzymes and kinase assay

*T. brucei* PKAR/PKAC1(TriTrypDB: Tb927.11.4610 and Tb927.9.11100); *T. cruzi* PKAR1/PKAC2 (TriTrypDB: TcCLB.506227.150 and TcCLB.508461.280), *L. donovani* PKAR1/PKAC1 (TritrypDB: LdBPK_130160.1 and LINF_350045600) isoform ORFs were amplified by PCR from their respective gDNA and fused to a 6xHis tag and a TEV cleavage site. PKACs were N-terminally fused to a strep tag. Mutations were introduced by PCR site directed mutagenesis via overlap extensions as described previously in *Ho et al., 1989*. Primer sequences are available in . The fusion ORFs were cloned into third generation pLEXSY vectors: pLEXSY_I-ble3 for PKARs and pLEXSY_I-neo3 for PKACs (Jena Bioscience). Holoenzymes were reconstituted in vivo by co-expression in *L. tarentolae* T7-TR according to the Jena Bioscience manual. The holoenzyme complexes were isolated using a tandem affinity purification protocol: Ni-NTA followed by Streptactin matrix. *L. tarentolae* cells were detergent lysed by vortex homogenization in Ni-NTA binding buffer (50 mM NaH$_2$PO$_4$ pH 7.4, 150 mM NaCl, 20 mM imidazole, 0.5% Triton-X 100, Complete Mini EDTA-free protease inhibitor cocktail (Roche)). The soluble fraction was loaded onto a gravity flow Ni-NTA column. After washing with Ni-NTA wash buffer (50 mM NaH$_2$PO$_4$ pH 7.4, 150 mM NaCl, 40 mM imidazole) the protein complex was eluted in Ni-NTA elution buffer (50 mM NaH$_2$PO$_4$ pH 7.4, 150 mM NaCl, 250 mM imidazole). The eluate was immediately loaded onto the gravity flow Streptactin column, washed with Streptactin wash buffer (50 mM NaH$_2$PO$_4$ pH 7.4, 150 mM NaCl) and eluted with Streptactin elution buffer (50 mM NaH$_2$PO$_4$ pH 7.4, 150 mM NaCl, 2.5 mM desthiobiotin). All purification steps were carried out at 4 °C. The mammalian PKA holoenzyme: human RIα/mouse Cα was co-expressed and co-purified from *E. coli* strain APE304 as previously described in *Bachmaier et al., 2019*. The kinase assays with radiolabelled [γ32P] ATP were set up and performed as described in *Hastie et al., 2006*. Briefly, a 50 µl kinase reaction mix was prepared at 4 °C by the addition of 5 µl of 10 x reaction buffer (500 mM MOPS pH 7; 1 M NaCl; 10 mM EGTA; 10 mM DTT; 1 mg/ml BSA; 100 mM MgCl$_2$), 5 µl kemptide (1 mM), 5 µl purified PKA holoenzyme. A test run using known activating ligands (*Bachmaier et al., 2019*) was carried out

and the kinase would then be diluted in 1 x reaction buffer, in order to work within the linear range of the assay. The Ligands were diluted in 30 µl $H_2O$ prior to addition. After temperature equilibration to 30 °C, the kinase reaction was started by addition of 5 µl 1 mM ATP spiked with [γ 32 P] ATP to give 200–400 cpm/pmole. The reaction was stopped after 10 min by pipetting 40 µl onto a 2x2 cm p81-phosphocellulose paper and immediate immersion into 75 mM phosphoric acid. Measurements were carried out in triplicates. Dose response curve fitting was performed with Graphpad prism's non-linear regression for calculation of half activation constants $EC_{50}$ and 95% confidence interval.

## Protein expression and purification for ligand binding studies

N-terminally truncated *T. brucei* PKAR(199-499) and *T. cruzi* PKAR(200-503) were cloned into pETDuet1 (Novagen) with a N-terminal 6xHis-tag. TbPKAR(199-499) mutants 6 and 7 (for sequences see *Table 2*) were generated by site-directed mutagenesis and cloned into pETDuet1. N-terminally truncated *L. donovani* PKAR(200-502) was fused to a Sumo_Ubiquitin Tag in a pET_Sumo vector (ThermoFisher). Refolded (RfAPO) and native (N) protein samples were subjected to nano differential scanning fluorimetry (nanoDSF), whereas for isothermal titration calorimetry (ITC), only refolded protein was used. Native and refolded proteins were prepared as reported by *Bachmaier et al., 2019* with the following modifications: native protein (N) eluted from affinity chromatography (Ni-NTA column) was dialyzed overnight and then directly probed for thermal stability using nanoDSF. Purification of LdPKAR1(200–502) by Ni-NTA affinity chromatography was followed by SenP2 protease mediated cleavage of the N-terminal Sumo tag during dialysis of the protein in 50 mM Hepes pH 7.5 and 50 mM NaCl (buffer B). After denaturation of TbPKAR(199-499) mutants 6 and 7, refolding occurred in a dialysis bag in presence of 1 mM cAMP or 1 mM cIMP, respectively. To mutant 7, additionally 6.5 moles of cIMP per mole of refolded protein were added before ITC measurements. Final elution of all proteins after Size Exclusion Chromatography (SEC) was in either 50 mM HEPES pH 7.5, 50 mM NaCl, and 1% DMSO (buffer A) or 50 mM HEPES pH 7.5 and 50 mM NaCl (buffer B). Preparation of cAMP-free human PKARIα was performed according to *Buechler et al., 1993*.

## Isothermal titration calorimetry (ITC)

ITC measurements were carried out on a MicroCal PEAQ-ITC (Malvern) instrument. Refolded proteins were diluted to 10–20 µM in buffer A or B. A total of 100–200 µM ligand were diluted in the same buffer as the protein and DMSO concentration of protein and ligand samples was adjusted as accurately as possible. As we observed that the molar ratio N decreased with time after final purification, we did all ITC experiments within a day to avoid precipitation of the protein. 2–4 µl of ligand were injected in a series of 13–19 injections into the protein sample at 298 K. The Differential Power (DP) between reference and sample cell was maintained at 8–10 µcal s$^{-1}$ in all experiments. Data analysis was performed with the MicroCal PEAQ-ITC software applying a model with one binding site.

## Thermal shift analysis using Nano Differentiation Scanning Fluorimetry (nanoDSF)

For nano differential scanning fluorimetry (nanoDSF), a Prometheus NT.48 (Nanotemper Technologies, Munich) equipped with high sensitivity glass capillaries (PR-C006, Nanotemper) was used. The technique allows label-free monitoring of protein melting temperatures (Tm). Upon heating 10 µl of protein sample per capillary from 15° to 90°, at a rate of 1–2°C per minute, intrinsic fluorescence at 330 and 350 nm ($F_{330}$/$F_{350}$) was recorded and the ratio of both or only the fluorescence at 330 nm was plotted as function of temperature. The melting temperature ($T_m$) was calculated from the first derivative of the curve, using the instrument's built-in software. Native (N) and refolded (RfAPO) protein preparations of TbPKAR(199-499), TbPKAR(199-499) mutant 6 and 7, and LdPKAR1(200–502) were subjected to nanoDSF before and after incubation in 1 mM of ligand(s). Accurate protein refolding was assumed when the melting temperature ($T_m$) of refolded and native samples, both loaded with an excess of 1 mM ligand, presented close matching values.

## Circular dichroism spectroscopy

For circular dichroism (CD) spectroscopy, native and refolded TbPKAR(199-499) samples from mutant 6 were prepared as described above with the following modifications: Refolding occurred in presence of 1 mM inosine and 1 mM cAMP, followed by a SEC on a Superdex 200 Increase

10/20 GL column (GE Healthcare) and elution in CD buffer (20 mM NaPi), free of chlorine. To ensure complete buffer change, the refolded protein was subsequently passed over a PD10 column (Ge Healthcare) and again eluted in CD buffer. Similarly, native protein was passed twice over PD10 columns with elution in CD buffer. Proteins were diluted to 2 μM (0.1mAU absorbance) and incubated with 10 μM inosine and 10 μM cAMP before measurement. The CD spectra were recorded using a Jasco J-815–150 S spectropolarimeter (Jasco, Tokyo, Japan) connected to a PTC 343 peltier set up to maintain the system at a constant temperature of 20 °C. The protein (sample volume = in 200 μL) was inserted into a rectangular quartz cell of 0.1 cm path length and the UV spectra recorded by averaging 20 scans in the wavelength of 185–260 nm. The CD signal was recorded in a window of –7–10 mdeg. The identification of the peaks in the spectra related to α-helices (193 nm) and ß-sheet (208 and 222 nm) enrichment were performed according to *Greenfield and Fasman, 1969*.

## Crystallization, X-ray diffraction data collection and structure determination of ligand-bound PKARs

Protein purification for crystallization of TbPKAR(199-499), TbPKAR(199-499) mutant 6 and TcPKAR(200-503) was performed as described in *Bachmaier et al., 2019* with the following modifications: Native protein eluted from a Ni-NTA column was cleaved by TEV protease for removal of the N-terminal 6xHis tag and then subjected to SEC. Protein freshly eluted from the Superdex 200 10/300 GL column was concentrated to at least 10 mg ml$^{-1}$ and, in order to ensure homogeneous ligand binding, incubated with either 1 mM inosine (TbPKAR(199-499) and TcPKAR(200-503)) or 1 mM cAMP (TbPKAR(199-499) mutant 6). Crystals grew within 7–10 days using sitting drops (100–500 nL) crystalizing via the vapor diffusion method (*Davies and Segal, 1971*). Crystals of TbPKAR(199-499) were obtained in 50 mM Tris pH 8.0, 4% MPD, 0.2 M ammonium sulfate, 32% PEG 3350 at 4 °C. Crystals of TcPKAR(200-503) were obtained in 20% PEG 3350, 0.2 M Magnesium acetate at 4 °C. Crystals of TbPKAR(199-499) mutant 6 were obtained in 50 mM Tris pH 8.0, 0.2 mM Magnesium Chloride, 30% PEG 3350 at 4 °C. Prior to flash cooling in liquid nitrogen, the crystals were briefly soaked in a mother liquor solution made of the reservoir buffer and 40% (v/v) of ethylene glycol. The X-ray diffraction data were collected at the Swiss Light Source beamline PXIII and on a Bruker D8 venture Metaljet system, at 100 K. The collected data were processed using XDS and scaled using XSCALE (*Kabsch, 2010*; *Kabsch, 2012*). The structure of TbPKAR(199-499) with inosine was solved using the Sulphur SAD (Single-wavelength Anomalous Diffraction) phasing method (*Doutch et al., 2012*). All other structures were solved by Molecular Replacement (MR) (*McCoy, 2007*) using the structure of TbPKAR +inosine as a search model in the software Phaser as implemented in PHENIX (*Liebschner et al., 2019*; *Adams et al., 2010*). All the MR solutions presented a TFZ score (Translation Function Z-score) >8 indicating correct solutions. The molecular models of the proteins were built using the 2Fo-Fc electron density map while the ligands were built using the difference map. The final structure was achieved by iterative cycles of manual building in Coot (*Emsley et al., 2010*) and refinement using PHENIX. Data collection and refinement statistics are summarized in .

## In silico docking of nucleosides to TbPKAR

In silico docking was performed using the software Glide (*Friesner et al., 2004*) as implemented in Maestro (Schrödinger). Ligands were built manually and prepared using LigPrep (Schrödinger). Ionization states and tautomers were not considered during ligand preparation. Stereoisomers had their chirality determined from the 3D structure (maximum 32 per ligand). The docking mode chosen was SP (Standard Precision). Chain B of TbPKAR (PDB: 6FLO) was chosen as a template for docking, since it presented a better overall electron density. For docking of inosine and guanosine to the A-site of TbPKAR the grid constrains used were E311 and water 96 (Match at least = 2). For docking of adenosine, the K293 conformer was changed to reach a hydrogen bond with the N7 of the purine ring. The grid restrains used were E311, K293 (Match at least = 2). For docking to the B-site, the chosen grid constraints were A444(N), G432(N), E435 and water 273 (Match at least = 2) for all three nucleosides. Poses were analysed by visual inspection and ranked according to Glide G-score (GG), a mathematical prediction of Gibbs Free Energy.

## Mass spectrometry analysis of ligands bound to TbPKAR in vivo

The TbPKAR ORF was N-terminally fused to a 6xHis tag by PCR and cloned into the pLEW82 expression vector and transfected into MITat1.2SM (single marker) blood stream form (BSF) cells and EATRO11252T7 insect stage form (PCF) cells, both of which expressed T7 polymerase and tetracycline repressor. Cell culture was exactly as reported before for BSF (*Bachmaier et al., 2019*) and PCF (*Schenk et al., 2021*). Transfected MiTat1.2SM blood stream forms were kept under constant selection with 1 µg/ml G418 and 2.5 µg/ml Bleomycin. Transfected EATRO11252T7 PCF cells were cultured under constant selection with 10 µg/ml G418, 25 µg/ml hygromycin and 2.5 µg/ml bleomycin. Selected clones were induced with 1 µg/ml tetracycline for 24 hours. The cells were harvested by centrifugation (1400 × *g* for 10 min), washed once with PBS and then detergent lysed in Ni-NTA binding buffer. The soluble fraction was incubated with magnetic Ni-NTA beads for 1 hr, followed by quick single washes in Ni-NTA binding buffer, Ni-NTA wash buffer, Streptactin wash buffer (50 mM $NaH_2PO_4$ pH 7.4, 150 mM NaCl) and finally MS-Grade $H_2O$. The beads were suspended in MS-grade water and boiled at 95 °C for 5 min. Beads and denatured protein precipitate were removed by centrifugation at 10,000 × *g* for 10 min. The supernatant was then transferred to a fresh tube and stored at –20 °C until analysis. For LC-ESI-MS, the samples were chromatographed by a Dionex Ultimate 3000 HPLC system with a flow of 0.15 ml/min over an Interchim Uptisphere 120 Å 3HDO C18 column (150x2 mm), while maintaining the column temperature at 30 °C. Elution was performed with buffer A (2 mM $HCOONH_4$ in $H_2O$, pH 5.5) and buffer B (2 mM $HCOONH_4$ in H2O/MeCN 20/80, pH 5.5), with a linear gradient from 0% to 15% buffer B in 45 min. The elution was monitored at 260 nm (Dionex Ultimate 3000 Diode Array Detector). The chromatographic eluent was directly injected into the ion source of a Thermo Finnigan LTQ Orbitrap XL without prior splitting. Ions were scanned by use of a positive polarity mode over a full-scan range of m/z 80–500 with a resolution of 30000. Parameters of the mass spectrometer were tuned with a freshly mixed aqueous solution of inosine (5 µM). The synthetic $^{13}C_5$-labeled internal isotope standards with an isotope enrichment of >99% were procured from Omicron Biochemicals Inc The quantification of nucleosides was carried out, as described in *Traube et al., 2019*, with the following amounts of the corresponding isotope labelled internal standards: 256.8 fmol $[^{13}C_5]$-inosine, 152.8 fmol $[^{13}C_5]$-guanosine, 662.8 fmol $[^{13}C_5]$-adenosine.

## Acknowledgements

We thank Thomas Carell, LMU Chemistry, for generous support of SB, MS instrument time and discussions, Ralph Heermann, LMU Microbiology, and Michaela Smolle, LMU BMC, for advice, and Andreas Anger, LMU Gene Center for structural homology modelling in the early phase of the project. Eleni Polatoglou launched YVS in the laboratory. Access to nanoDSF instruments and advice was generously provided by NanoTemper Technologies GmbH (Munich, Germany). We are grateful to Oliver Plettenburg for discussions and Ricardo Biondi, IBIOBA, Buenos Aires for critical reading of the manuscript. The work was supported by the Bundesministerium für Bildung und Forschung (BMBF) grant 16GW0281-3 to MB and FS. YVS was supported by a fellowship from the Brazilian Science Without Borders/CNPq program and by the Life Sciences Munich (LSM) graduate school.

## Additional information

### Competing interests

Frank Schwede: CEO/CSO of BIOLOG Life Science Institute GmbH & Co. KG. The other authors declare that no competing interests exist.

### Funding

| Funder | Grant reference number | Author |
|---|---|---|
| Bundesministerium für Bildung und Forschung | grant reference 16GW0281-3 | Frank Schwede Michael Boshart |

| Funder | Grant reference number | Author |
| --- | --- | --- |
| Conselho Nacional de Desenvolvimento Científico e Tecnológico | PhD fellowship | Yuri Volpato Santos |
| Deutsche Forschungsgemeinschaft | BO1100/7-1 | Michael Boshart |
| Brazilian Science Without Borders/CNPq program | | Yuri Volpato Santos |
| Life Sciences Munich Graduate School | | Yuri Volpato Santos |

The funders had no role in study design, data collection and interpretation, or the decision to submit the work for publication.

## Author contributions
Veronica Teresa Ober, Formal analysis, Validation, Investigation, Visualization, Writing - original draft, Writing - review and editing; George Boniface Githure, Conceptualization, Formal analysis, Investigation, Visualization, Methodology, Writing - original draft; Yuri Volpato Santos, Conceptualization, Formal analysis, Investigation, Visualization, Methodology, Writing - original draft, Writing - review and editing; Sidney Becker, Formal analysis, Validation, Investigation, Visualization, Methodology; Gabriel Moya Munoz, Investigation; Jérôme Basquin, Data curation, Formal analysis, Validation, Methodology; Frank Schwede, Resources; Esben Lorentzen, Supervision, Validation; Michael Boshart, Conceptualization, Formal analysis, Supervision, Funding acquisition, Validation, Visualization, Writing - original draft, Project administration, Writing - review and editing

## Author ORCIDs
Veronica Teresa Ober  http://orcid.org/0000-0001-8631-9255
Yuri Volpato Santos  http://orcid.org/0000-0002-9545-8430
Sidney Becker  http://orcid.org/0000-0002-4746-2822
Esben Lorentzen  http://orcid.org/0000-0001-6493-7220
Michael Boshart  http://orcid.org/0000-0002-5070-2663

Reviewer #1 (Public Review): https://doi.org/10.7554/eLife.91040.3.sa1
Reviewer #2 (Public Review): https://doi.org/10.7554/eLife.91040.3.sa2
Author Response https://doi.org/10.7554/eLife.91040.3.sa3

# Additional files

## Supplementary files
• Supplementary file 1. Binding parameters from ITC measurements.

• Supplementary file 2. Data collection and refinement statistics for the crystal structures.

• Supplementary file 3. Results of searching the mass spectrometry data set (*Figure 6—figure supplement 1*) for nucleosides matches in the MODOMICS (https://iimcb.genesilico.pl/modomics/) RNA modifications database. Red cross: peak/mass not detected. Red: peak Rf/mass: detected but not significant over background. Green mass: peak Rf/mass: detected but not confirmed.

• Supplementary file 4. List of primers used in this study.

• MDAR checklist

## Data availability
The coordinates of the crystal structures of *T. cruzi* PKAR bound to inosine, *T. brucei* PKAR bound to inosine and *T. brucei* PKAR (mutant 6) bound to cAMP and inosine have been deposited in the Protein Data Bank under the accession codes 6HYI, 6FLO, 6H4G, respectively. Genome sequence and annotation information was obtained from TritrypDB (http://www.tritrypdb.org). Results from a search of the MODOMICS database (Boccaletto et al. 2022) to identify nucleoside analogues identified in living organisms is provided as *Supplementary file 1*.The source data underlying figures, tables, and figure supplements are provided as source data files.

The following datasets were generated:

| Author(s) | Year | Dataset title | Dataset URL | Database and Identifier |
|---|---|---|---|---|
| Volpato Santos Y, Lorentzen E, Basquin J, Boshart M | 2019 | Regulatory subunit of a cAMP-independent protein kinase A from *Trypanosoma brucei* at 2.1 Angstrom resolution | https://doi.org/10.2210/pdb6FLO/pdb | Worldwide Protein Data Bank, 10.2210/pdb6FLO/pdb |
| Volpato Santos Y, Lorentzen E, Basquin J, Boshart M | 2019 | Regulatory subunit of a cAMP-independent protein kinase A from *Trypanosoma cruzi* at 1.4 A resolution in complex with inosine | https://doi.org/10.2210/pdb6HYI/pdb | Worldwide Protein Data Bank, 10.2210/pdb6HYI/pdb |
| Volpato Santos Y, Lorentzen E, Basquin J, Boshart M | 2019 | Regulatory subunit of a cAMP-independent protein kinase A from *Trypanosoma brucei*: E311A, T318R, V319A mutant bound to cAMP in the A site | https://doi.org/10.2210/pdb6H4G/pdb | Worldwide Protein Data Bank, 10.2210/pdb6H4G/pdb |

The following previously published dataset was used:

| Author(s) | Year | Dataset title | Dataset URL | Database and Identifier |
|---|---|---|---|---|
| Su Y, Dostmann WRG, Herberg FW, Durick K, Xuong N-H, Ten Eyck L, Taylor SS, Varughese KI | 1996 | REGULATORY SUBUNIT OF CAMP DEPENDENT PROTEIN KINASE | https://www.rcsb.org/structure/1RGS | RCSB Protein Data Bank, 1RGS |

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
