## [Editor Report · eLife assessment]

This **landmark** study sheds light on a long-standing puzzle in Protein kinase A activation in Trypanosoma. Extensive experimental work provides **exceptional** evidence for the conclusions of the work, which represents a significant advancement in our understanding of the molecular mechanism of cyclic nucleotide binding domains. The work is relevant for researchers with interests in kinases and their mechanistic study.

---

## [Referee Report · Reviewer #1 (Public Review)]

Summary:

Cyclic Nucleotide Binding (CNB) domains are pervasive structural components involved in signaling pathways across eukaryotes and prokaryotes. Despite their similar structures, CNB domains exhibit distinct ligand-sensing capabilities. The manuscript offers a thorough and convincing investigation that clarifies numerous puzzling aspects of nucleotide binding in Trypanosoma.

---

## [Referee Report · Reviewer #2 (Public Review)]

Summary:

This manuscript clearly shows that Trypanosoma PKA is controlled by nucleoside analogues rather than cyclic nucleotides, which are the primary allosteric effectors of human PKA and PKG. The authors demonstrate that the inosine, guanosine, and adenosine nucleosides bind with high affinity and activate PKA in the tropical pathogens *T. brucei*, *T. cruzi* and Leishmania. The underlying determinants of nucleoside binding and selectivity are dissected by solving the crystal structure of T. cruzi PKAR(200-503) and *T. brucei* PKAR(199-499) bound to inosine at 1.4 Å and 2.1 Å resolution and through comparative mutational analyses. Of particular interest is the identification of a minimal subset of 2-3 residues that controls nucleoside vs. cyclic nucleotide specificity.

---

## [Author Response]

The following is the authors’ response to the original reviews.

**eLife assessment**
This landmark study sheds light on a long-standing puzzle of Protein kinase A activation in Trypanosoma. Extensive experimental work provides compelling evidence for the conclusions of the manuscript. It represents a significant advancement in our understanding of the molecular mechanism of Cyclic Nucleotide Binding domains and will be of interest to researchers with interest in kinases and mechanistic studies.**Public Reviews**:
**Reviewer #1 (Public Review):**
Summary:Cyclic Nucleotide Binding (CNB) domains are pervasive structural components involved in signaling pathways across eukaryotes and prokaryotes. Despite their similar structures, CNB domains exhibit distinct ligand-sensing capabilities. The manuscript offers a thorough and convincing investigation that clarifies numerous puzzling aspects of nucleotide binding in Trypanosoma.Strengths:One of the strengths of this study is its multifaceted methodology, which includes a range of techniques including crystallography, ITC (Isothermal Titration Calorimetry), fluorimetry, CD (Circular Dichroism) spectroscopy, mass spectrometry, and computational analysis. This interdisciplinary approach not only enhances the depth of the investigation but also offers a robust cross-validation of the results.Weaknesses:None noticed.
**Reviewer #2 (Public Review):**
Summary:This manuscript clearly shows that Trypanosoma PKA is controlled by nucleoside analogues rather than cyclic nucleotides, which are the primary allosteric effectors of human PKA and PKG. The authors demonstrate that the inosine, guanosine, and adenosine nucleosides bind with high affinity and activate PKA in the tropical pathogens *T. brucei*, *T. cruzi* and Leishmania. The underlying determinants of nucleoside binding and selectivity are dissected by solving the crystal structure of *T. cruzi* PKAR(200-503) and *T. brucei* PKAR(199-499) bound to inosine at 1.4 Å and 2.1 Å resolution and through comparative mutational analyses. Of particular interest is the identification of a minimal subset of 2-3 residues that controls nucleoside vs. cyclic nucleotide specificity.Strengths:The significance of this study lies not only in the structure-activity relationships revealed for important targets in several parasite pathogens but also in the understanding of CNB's evolutionary role.Weaknesses:The main missing piece is the model for activation of the kinetoplastid PKA which remains speculative in the absence of a structure for the trypanosomatid PKA holoenzyme complex. However, this appears to be beyond the scope of this manuscript, which is already quite dense.

We fully agree that insight into the activation mechanism and its possible deviation from the mammalian paradigm requires a holoenzyme structure revealing the details of R-C interaction. We have attempted Cryo-EM from LEXSY-produced holoenzyme, yet upscaling the purification procedures described in this manuscript have repeatedly failed in spite of numerous protocol changes and optimizations. Much more work is required to achieve this.

**Reviewer #2 (Recommendations For The Authors):**
Some minor points to consider for enhancing the impact of this interesting manuscript:(1) The nucleoside affinities measured are mainly for the regulatory subunits unbound to the kinase domain. How would nucleoside affinities change when the regulatory subunits are bound to the kinase domain, which is presumably the case under resting conditions? An estimation of this change in affinity is important because it more closely relates to the variations in cellular nucleoside concentrations needed for activation.

This is an important question and we have given an indirect answer in the manuscript, but not very explicit. The EC50 values for kinase activation of the purified holoenzyme complexes are very similar or almost identical to the kD values measured by ITC with free regulatory subunits. By inference, the binding kD for the holoenzyme and for the free R-subunit cannot be very different. In addition, we have recently determined the EC50 for PKA activation in vivo in trypanosomes using a bioluminescence complementation reporter assay. The values fit perfectly to the values obtained with purified holoenzyme (Wu et al. in preparation). A sentence in Results (lines 201-203) has been added.

(2) The authors should point out that a major implication of nucleoside vs. cyclic nucleotide activation is in terms of signal termination. If phosphodiesterases (PDEs) are responsible for cAMP/cGMP signal termination, what terminates nucleoside-dependent signaling? Although the answer to this question may not be known at this stage, it is important to highlight this critical implication of the authors' study.

The mechanism of signal termination is indeed unknown so far. We speculate that some enzymes of the purine salvage pathways are differentially localized in subcellular compartments and thereby able to establish microdomains that enable nucleoside signaling. In addition, PKA subunit phosphorylations/dephosphorylations and/or protein turnover may also regulate signal termination. As an example, free PKAC1 is rapidly degraded upon depletion of the PKAR subunit by RNAi. We have now mentioned signal termination in Discussion and have revised the last part of Discussion (lines 567-602). A possible approach to monitor compartmentalized signaling would be using the FluoSTEPs technology (Tenner et al., Sci. Adv. 2021; 7: eabe4091), but adapting this to the trypanosome system will not be a short-term task.